# Hybrid metagenome assemblies link carbohydrate structure with function in the human gut microbiome

Anuradha Ravi[1,4,6], Perla Troncoso-Rey[1,6], Jennifer Ahn-Jarvis[1,6], Kendall R. Corbin[1,2], Suzanne Harris[1,5], Hannah Harris[1], Alp Aydin[1], Gemma L. Kay[1], Thanh Le Viet[1], Rachel Gilroy[1], Mark J. Pallen[1], Andrew J. Page[1], Justin O'Grady [1,3,6] & Frederick J. Warren [1,6✉]

Complex carbohydrates that escape small intestinal digestion, are broken down in the large intestine by enzymes encoded by the gut microbiome. This is a symbiotic relationship between microbes and host, resulting in metabolic products that influence host health and are exploited by other microbes. However, the role of carbohydrate structure in directing microbiota community composition and the succession of carbohydrate-degrading microbes, is not fully understood. In this study we evaluate species-level compositional variation within a single microbiome in response to six structurally distinct carbohydrates in a controlled model gut using hybrid metagenome assemblies. We identified 509 high-quality metagenome-assembled genomes (MAGs) belonging to ten bacterial classes and 28 bacterial families. Bacterial species identified as carrying genes encoding starch binding modules increased in abundance in response to starches. The use of hybrid metagenomics has allowed identification of several uncultured species with the functional potential to degrade starch substrates for future study.

[1] Quadram Institute Bioscience, Norwich Research Park, Norwich NR4 7UQ, UK. [2] Department of Horticulture, University of Kentucky, Lexington, KY, USA. [3] University of East Anglia, Norwich Research Park, Norwich NR4 7TJ, UK. [4] Present address: Gemini centre for Sepsis Research, Department of Circulation and Medical Imaging, Norwegian University of Science and Technology, Trondheim, Norway. [5] Present address: London School of Hygiene & Tropical Medicine, Keppel Street, London WC1E 7HT, UK. [6] These authors contributed equally: Anuradha Ravi, Perla Troncoso-Rey, Jennifer Ahn-Jarvis, Justin O'Grady, Frederick J. Warren. ✉email: fred.warren@quadram.ac.uk

Microbial diversity within the microbiome and its interactions with host health and nutrition are now widely studied[1]. An important role of the human gut microbiome is the metabolic breakdown of complex carbohydrates derived from plants and animals (e.g. legumes, seeds, tissue and cartilage)[2]. Short-chain fatty acids are the main products of carbohydrate fermentation by gut microbiota and provide a myriad of health benefits through their systemic effects on host metabolism[3,4]. However, we still do not have a complete picture of the range of microbial species involved in the fermentation of complex carbohydrates to produce short-chain fatty acids. Understanding the intricacies of complex carbohydrate metabolism by the gut microbiota is a substantial challenge. The function of many 'hard to culture species' remains obscure and while advances in sequencing technology are beginning to reveal the true diversity of the human gut microbiota, there is still much to be learned[5].

A key challenge is understanding the influence of the structural complexity of carbohydrates on microbiota composition. Carbohydrates possess immense structural diversity, both at the chemical composition level (monomer and sugar linkage composition) and at the mesoscale. Individual species, or groups of species, within the gut microbiota are highly adapted to defined carbohydrate structures[6]. Starch is representative of the structural diversity found amongst carbohydrates and serves as a good model system as starches are readily fermented by several different species of colonic bacteria[7]. The gut microbiota is repeatedly presented with starches of diverse structures from the diet[8]. Consistent in starch is an $\alpha\text{-}1 \rightarrow 4$ linked glucose backbone, interspersed with $\alpha\text{-}1 \rightarrow 6$ linked branch points. Despite this apparent structural simplicity, starches botanical origin and subsequent processing (e.g. cooking) impact its physicochemical properties, particularly crystallinity and recalcitrance to digestion[7]. It has been shown in vitro[7], in animal models[9] and in human interventions[8], that altering starch structure can have a profound impact on gut microbiome composition.

The microbiome is known to harbour a huge repertoire of carbohydrate-active enzymes (CAZymes) that can degrade diverse carbohydrate structures[10,11]. However, it is a formidable challenge to study this functionality in complex microbial communities due to limitations in the depth of sequencing and coverage of all members in the community. While metagenomic sequencing has become a key tool, identifying genomes and functional pathways within the microbiome remains challenging in second-generation sequencing due to limitations associated with short (~300 bp) reads. Third-generation sequencing such as nanopore sequencing (ONT), promises to circumvent these difficulties by providing longer reads (>3 kilobase pairs [kbp]). This technology has become popular in clinical metagenomics for rapid pathogen diagnosis[12] and in human genomics research[13]. Long-read sequences can help bridge inter-genomic repeats and produce better de novo assembled genomes[14]. While the MinION platform from ONT has been used for metagenomic studies[15], it cannot provide sufficient sequencing depth and coverage to sequence the many hundreds of genomes present in the human gut microbiome. PromethION (ONT) can produce far greater numbers of sequences compared to either MinION or GridION, averaging four-five times more data per flow cell and the capacity to run up to 48 flow cells in parallel; this makes it suitable for metagenomics and microbiome studies. For example, PromethION has been used for long-read sequencing of environmental samples such as wastewater sludge, demonstrating its potential to recover large numbers of metagenome-assembled genomes (MAGs) from diverse mixtures of microbial species[16]. However, long error-prone reads aren't ideal for species resolution metagenomics, therefore, a hybrid approach using short and long-read data has been found to be most effective for generating accurate MAGs[17].

To achieve species-level resolution of the microbes present in the gut microbiome during complex carbohydrate utilisation, we conducted a genome-resolved metagenomics study in a controlled gut colon model. In vitro fermentation systems have been used extensively to model changes in the gut microbial community because of external inputs, e.g. changes in pH, protein, and carbohydrate supply[7,18,19]. We measured the dynamic changes in bacterial populations during fermentation of six structurally contrasting substrates: two highly recalcitrant starches (native Hylon VII (Hylon) and native potato starch (Potato)); two accessible starches (native normal maize starch (N. maize) and gelatinised then retrograded maize starch (R. maize)); an insoluble fibre (cellulose) resistant to fermentation (Avicel); and a highly fermentable soluble fibre (Inulin). By generating hybrid assemblies using PromethION and NovaSeq data, we obtained 509 MAGs. The dereplicated set consisted of 151 genomes belonging to ten bacterial classes and 28 bacterial families. Using genome-level information and read proportions data, we identified several species that have putative starch-degrading properties.

## Results

**PromethION and NovaSeq sequencing of model gut samples enriched for carbohydrate-degrading species.** Fermentation of six contrasting carbohydrate substrates (Inulin, Hylon, N. maize, Potato, R. maize and Avicel; see methods section) was initiated by inoculation of the model colon with a carbohydrate and faecal material and the gut microbial community composition was monitored over time (0, 6, 12 and 24 h) by sequencing as shown in Fig. 1. A more detailed workflow of the analyses is shown in Supplementary Fig. 1. In total, 23 samples and negative control were sequenced.

*PromethION sequencing.* The two sequencing runs generated 144 giga base pairs of raw sequences. In the first run, all 23 samples were analysed while in the second batch, 12 samples from Hylon, Inulin and R. maize were selected. The first run produced 7.87 million reads with an average read length of 3419 ± 57 bp and the second run generated 21.6 million reads with an average read length of 4707 ± 206 bp. Consolidating the runs, trimming and quality filtering resulted in the removal of 33.3 ± 14.7 % of reads (Supplementary Data 1). Median read lengths after trimming were 4972.5 ± 229 bp and the median quality score was 9.7 ± 0.9.

*Illumina sequencing.* All 23 samples provided high-quality sequences (Q value >30) generating a mean of 27 million reads per sample. Quality and read length (<60 bp) filtering removed 2.96% of reads (Supplementary Data 1).

**Dynamic shifts in taxonomic profiles among carbohydrate treatments.** Hierarchical clustering for the taxonomic profiling using MetaPhlAn3 for each sample is shown in Fig. 2 and Supplementary Data 2. At baseline (0 h), profiles of the top 30 selected species by clustering (using Bray-Curtis distances for samples and species, and a complete linkage) is similar for all treatments, as expected (Fig. 2a). This uniform profile was distinct from the water control sample (a.k.a. 'the kitome'). The water blank also had less than 3% (NovaSeq) and less than 0.2% (PromethION) reads compared to the other samples. The most abundant species in all the substrates was consistently *Prevotella copri* which decreased in abundance over time but remained one of the most abundant species throughout. *Faecalibacterium praunitzii* decreased in abundance in Inulin at 6 and 12 h and then increased in abundance for Inulin and Avicel at 24 h.

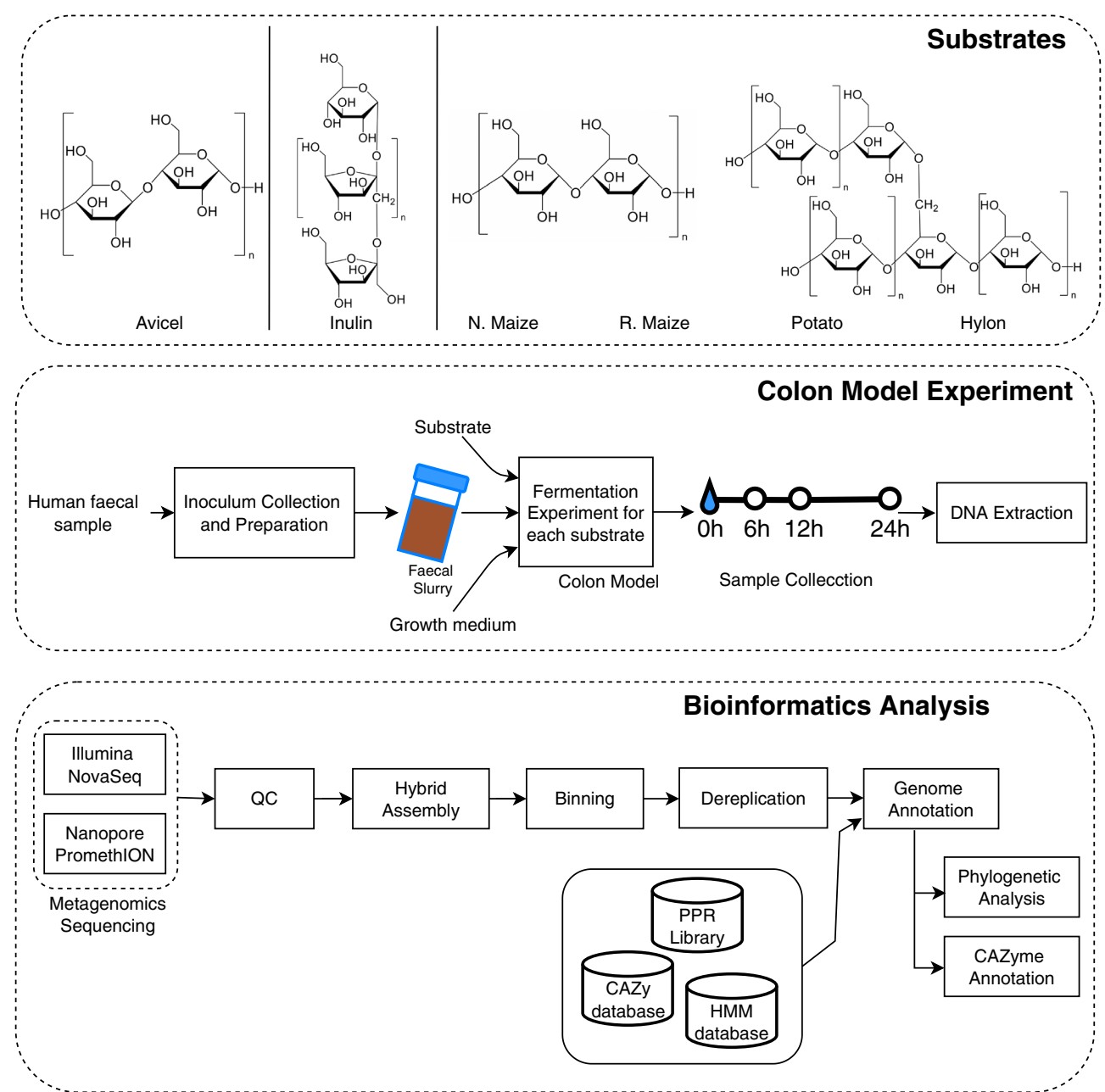

**Fig. 1 Workflow of the bioinformatic analyses.** Analysis of combined Illumina NovaSeq and Oxford Nanopore PromethION metagenomics data collected in a model colon study of the fermentation of different carbohydrate substrates with contrasting structures (Avicel, Inulin, Normal maize (N. maize), Retrograded maize (R. maize), Potato and Hylon) to study differences in the gut microbiota using a human stool sample.

Microbiome shifts were apparent from 6 h in the N. maize treatment which showed a very high abundance of *E. coli* (Fig. 2b and Supplementary Fig. 2). After 12 h, the profiles changed further with a higher abundance of *Escherichia coli* and *Bifidobacterium animalis* in the N. maize treatment. By the last sampling point (24 h), starch-specific species such as *E. coli, Ruminococcus bromii, Bifidobacterium adolescentis, B. animalis* showed an increase in abundance in Hylon, Potato and R. maize (Supplementary Fig. 2). *E.scherichia coli* showed the highest dominance in N. maize with a 17x greater abundance than at time 0 h.

Dynamic shifts in the microbiome were estimated using Principle Component Analysis. The distances between the microbiomes of the different treatments diverged significantly at 6 h (*p*.value: 0.029, Permdisp ANOVA), 12 h (*p*.value: 0.022,

Permdisp ANOVA) and at 24 h (*p*.value: 0.0099, Permdisp ANOVA). Within the time points, the profiles of R. maize and Inulin treatment profiles progressed similarly to each other, as did the Potato and Hylon treatment profiles (Supplementary Fig. 3). Avicell, an insoluble fibre, showed a distinct progression of microbiome, unlike the starch sources. The most distinct taxonomic change in microbial community composition in the Avicel treatment was apparent after 6 h. The microbiome of N. maize progressed differently from all other treatments with *E. coli* taking over most variance from 6 to 24 h. Figure 2b shows a PCoA plot of all the treatments, excluding samples treated with N. maize to allow smaller differences between the other samples to be seen. Inverse Simpson index results followed a similar pattern for changes in diversity for most treatments, which decreased at 6 h followed by a gradual increase at 12 and 24 h

a.

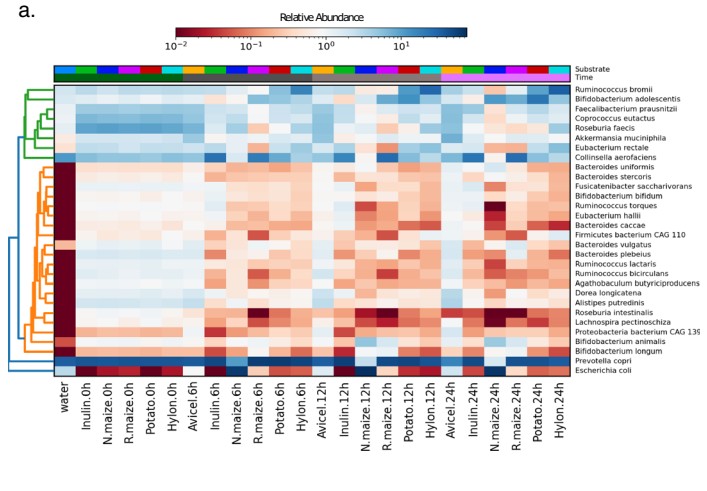

b.

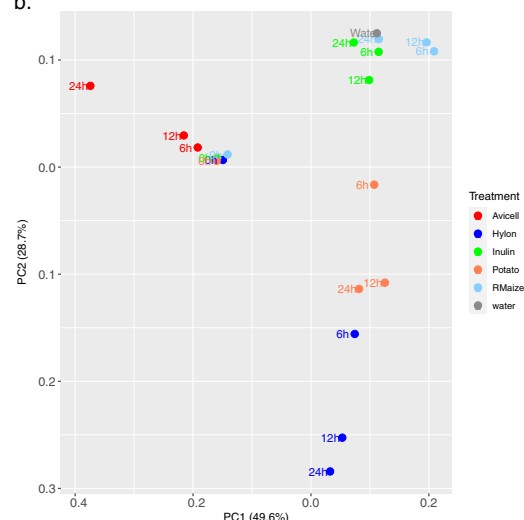

**Fig. 2 Hierarchical clustering and principle component analyses. a** Hierarchical clustering using Hclust2 of the top 30 selected gut microbial species present after fermentation of Avicel, Inulin, Normal maize (N.maize), Retrograded maize (R.maize), Potato and Hylon at 0, 6, 12, and 24 h in the model colon. The hierarchical clustering also includes a water sample (the kitome). **b** Principal component analysis (PCoA) excluding N.maize. PCoA plot showing dynamics of the microbiome during the different time points and between the carbohydrate treatments. Principle component (PC)1 and PC2 represent the percentage of variance explained by PC1 and 2.

(Supplementary Fig. 4). Avicel treatment showed a different pattern of taxonomic shifts with many taxa increasing in abundance at 12 h and 24 h. Empirical distances between the microbiomes of different treatments between time periods showed the microbiomes were very similar at time 0 as expected (distance to centroid at 0 h–0.023) (Supplementary Fig. 5).

Microbial metabolic gene pathways profiles were obtained using the HUMAnN3 tool (Supplementary Data 3). Dynamic shifts in microbial community that also reflects shifts in the abundance of metabolic gene pathways was quantified (Supplementary Data 4–9). These dynamic pathways and genes were defined as those showing at least a 1.5-fold (log2) shift in abundance relative to baseline (time 0 h).

With limited taxonomic shifts observed in Avicell treatment, very few genes showed distinct changes, mainly linked to cell wall remodelling and core microbial metabolism (Supplementary Data 10). There was a 1.5x shift from 0 to 6 h and a 1.2x shift from 0 to 24 h in an abundance of a 6-phospho-beta-glucosidase, which can breakdown products of cellulose, possibly indicating some very limited cellulose degradation. Inulin shows a broader range of enzymes involved in microbial cellular metabolism. A twofold increase in abundance of 1,5-anhydro-D-fructose reductase was observed, an enzyme potentially involved in the metabolism of products from inulin metabolism (Supplementary Data 11). N. maize shows a non-specific pattern of gene abundance due to the high levels of *E.coli* (Supplementary Data 12). The substrates Potato, Hylon and R. maize all show specific changes in gene abundances related to starch degradation (Supplementary Data 13–15). At 24 h, Hylon shows a twofold, and potato a 1.5-fold increase in alpha-amylase gene abundance. Potato (1.6-fold) and R. maize (1.5-fold) also show increased oligo-1,6-glucosidase involved in degrading the 1,6 branch points present in Potato and R. maize (but largely absent in Hylon). In addition to starch hydrolysing enzymes, this analysis also revealed increases in the abundance of a wide range of genes directly involved in acetyl-CoA and fatty acid metabolism for the Potato, R. maize and Hylon substrates. This may be linked to the production of high levels of short-chain fatty acids as an endpoint of the metabolism of these substrates.

**Hybrid metagenome assemblies vs short-read only assemblies.** Using Opera-MS, we combined PromethION reads with Illumina assemblies to produce hybrid assemblies. The assembly statistics for short-read-only and hybrid assemblies are shown in Supplementary Table 1 and Supplementary Fig. 6. The longest N50 and the largest contig per treatment were generated using hybrid assemblies (Supplementary Fig. 6b, c). The overall length of assembled sequences was similar for both approaches (Supplementary Fig. 6d).

The reads from each treatment and collective T0 were co-assembled into hybrid assemblies and binned into MAGs. In total, we binned and refined 509 MAGs that met the MIMAG quality score criteria[20] through checkM (Supplementary Data 16). Around 65% ($n = 331$) of the MAGs were high-quality with completeness of >90% and contamination <5% (Fig. 3a–c).

From the co-assemblies, thirty-five MAGs had an N50 of >500,000 Mbp and 158 MAGs were assembled into <30 scaffolds. The MAGs were dereplicated into primary and secondary clusters according to Average Nucleotide identity (ANI) (primary clusters <97%; secondary clusters <99%). In total, we identified 151 MAG secondary unique clusters (Supplementary Data 17). Each unique cluster consisted of between one and seven genomes based on their ANI threshold.

GTDb was used to propose bacterial taxonomy to the representative MAGs (Supplementary Data 17). All MAG clusters had >99% identity to existing genera (Supplementary Table 2). Here, 46 of the 151 MAG clusters were named using alphanumeric genus names and 19 MAGs clusters were named using alphanumeric species names. To provide clear and stable genus and species names, the MAGs were directly compared in NCBI to check whether these MAGs were already named or had any culture representatives. We identified 15 MAGs that were previously named and/or cultured so the Latin binomial for these MAGs was updated. For the rest of the MAGs, we used the approach described in Pallen et al.[21] to provide Latin names to 53 MAG clusters (Supplementary Data 18). We have provided 12 genus names and 51 species names (Supplementary Note 1) to genomes previously not identified in NCBI. In addition, MAG assembly statistics for the MAGs in the present study was

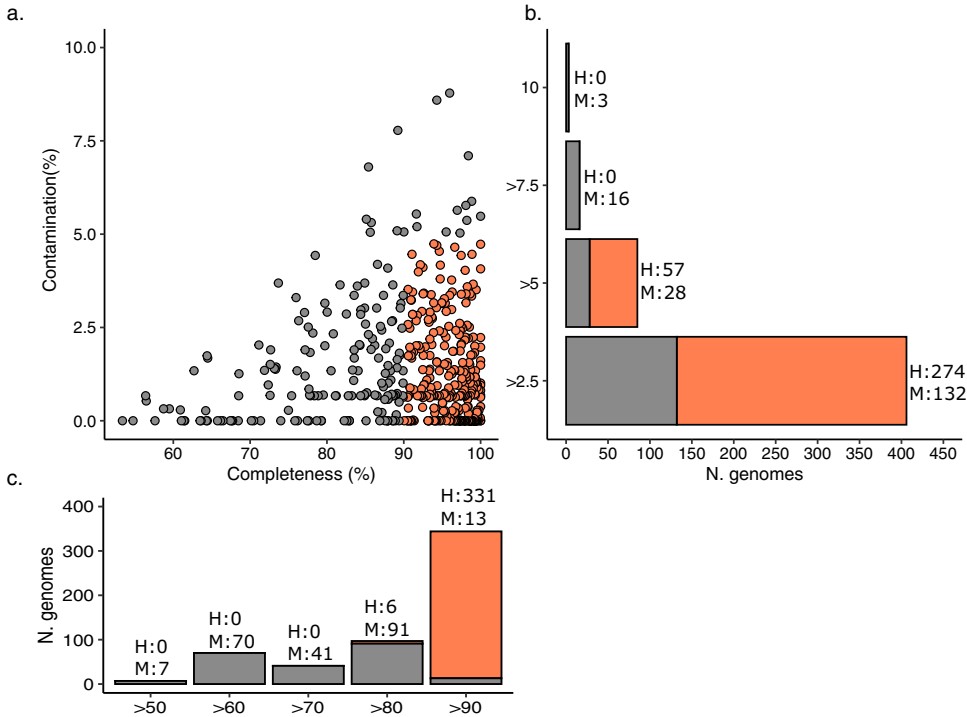

**Fig. 3 Assessment of metagenome-assembled genome (MAG) quality scores.** Completeness and contamination scores were estimated using CheckM. Colours are based on the MAG standards: orange- high quality (H) as >90% completeness and <5% contamination; Grey—Medium quality (M) as <90–60% completeness and >5–10% contamination. **a** The dots within the plot represent each MAG that passed quality standards. **b** Horizontal stacked bar plot representing the number of high and medium quality MAGs with different contamination scores. N. genomes denotes the number of genomes. H and M values represent the number of genomes. **c** Vertical stacked bar charts representing the number of high and medium quality MAGs with different completion scores. H and M values represent the number of genomes.

compared to the representative assemblies in GTDb (Supplementary Data 19). We found that while the average overall assembly length was almost similar (an average of 2,237,598 bp in the present study versus 2,533,373 bp in GTDb), there were far fewer contigs in our assemblies (an average of 68 contigs in the present study versus 162 in GTDb), and therefore our MAGs may be of higher quality than those present in reference databases.

**Carbohydrate structure drives progression of bacterial diversity.** Logarithmic fold change between time points of each MAG was calculated to estimate the changes in the relative abundance of the MAG between treatments (Supplementary Data 20). In total, 36 of 151 clusters exhibited ≥2x-log fold increase in relative abundance. Specifically, ≥2x-log fold change in abundance was seen in 6, 9, 12, 11 and 18 MAGs for Avicel, N. maize, Hylon, Potato and R. maize treatments, respectively. The genomes were partitioned as early (0 and 6 h) and late degraders (12 and 24 h) according to when they first showed an increase in relative abundance (Supplementary Table 3). We identified several genomes, to the best of our knowledge not previously identified as degraders of the different carbohydrate treatments (Fig. 4). Avicel is highly crystalline cellulose that is resistant to fermentation; the human gut microbiota has a very limited capacity to degrade celluloses[22]. Interestingly, the largest increase in abundance we observed was for *Blautia hydrogenotrophica*; which has been reported in association with cellulose fermentation since it acts as an acetogen using hydrogen produced by primary degraders of cellulose[23]. *Bacteroides uniformis* has been previously characterised as an inulin-degrading species[24], and in our analysis, it was identified to increase from 12 h during inulin fermentation. In addition to this, *Faecalibacterium prausnitzii* increased in abundance with inulin supplementation and has been shown to

have the ability to degrade inulin when co-cultured with primary degrading species[25,26]. Our analysis identified several well-known starch-degrading species, most notably *R. bromii* (Fig. 4). *R. bromii* was identified in the most recalcitrant starch treatments i.e., Hylon, Potato and R. maize treatments. Maize starch treatments (R. maize, N. maize and Hylon) showed increases in abundance of *Bifidobacterium* species. Previous studies have characterised *Bifidobacterium* as a starch-degrading genus[27]. The only *Bifidobacterium* species to increase in abundance in response to Hylon was *B. adolescentis*, which is known to utilise this hard-to-digest starch better than other *Bifidobacterium* species[28]; a broader range of *Bifidobacterium* species increased in abundance in response to the more accessible R. maize and N. maize substrates, suggesting these species may be better adapted to more accessible starches. In the Potato treatment, another closely related but less well-characterised *Rumminococcus* species, *Candidatus Ruminococcus anthropic* was identified.

Aggregation of relative abundance from each treatment showed the activity of the microbiome between each time point at a MAG level (Supplementary Fig. 7). Relative abundance was constant for Avicel throughout, indicating low activity of the MAGs in utilising crystalline cellulose, likely reflecting the very limited fermentability of microcrystalline cellulose. As for other maize starches (Hylon, R. maize and N. maize), the read proportions showed an overall reduction in abundance, with only starch-degrading MAGs increasing in abundance.

**CAZyme family interplay with the carbohydrate treatments and organisation into polysaccharide utilisation loci.** To identify CAZymes in the MAGs, genome-predicted proteins identified by Prodigal were compared with the CAZy database using dbCAN2 (Supplementary Data 21, Supplementary Fig. 8).

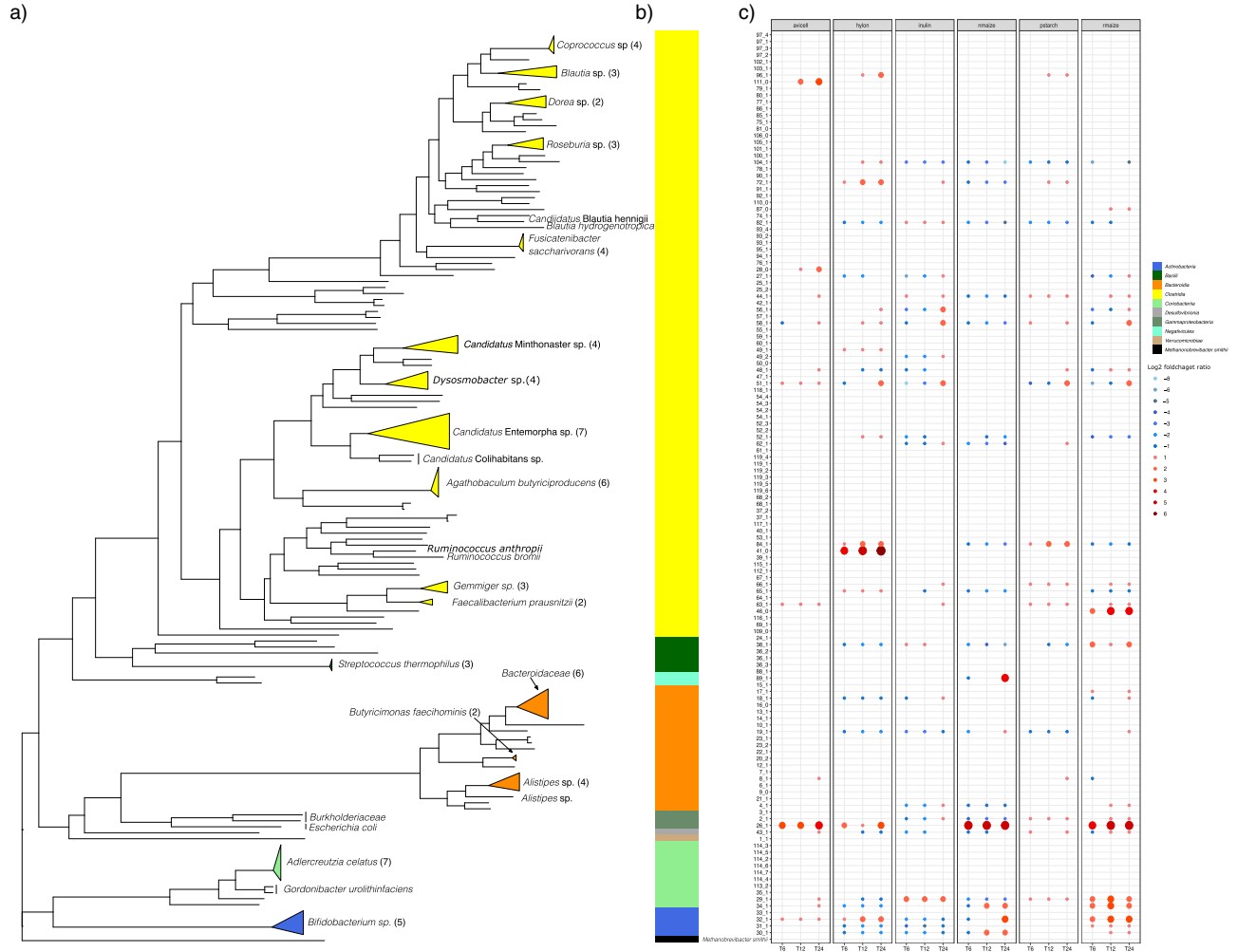

**Fig. 4 Phylogenomic tree of the MAGS and log fold changes in read proportions between the different starch treatments. a** The phylogenetic tree was constructed from concatenated protein sequences using PhyloPhlAn and illustrated using ggtree. Clades belonging to similar bacterial family and bacterial genus were collapsed. **b** The colour strip represents the phylum-level distribution of the phylogenetic tree. **c** Dot plot shows the decrease (negative log$_2$ fold change; blue shades) and increase (positive log$_2$ fold change; red shades) of read proportions from 0 to 6 h, 0 to 12 h and 0 to 24 h for all treatments.

We further analysed the genome organisation of the CAZymes identified by dbCAN2 using the tool PULpy[29], which identified Polysaccharide Utilisation Loci (PULs) in a total of 21 MAGs (Supplementary Data 22). All the PULs identified were within the phylum *Bacteroidetes*, and the most PULs identified within a single MAG was 79 found in *Butyricimonas faecihominis*. These statistics for the numbers of PULs identified are comparable to other studies published using the same tool[17]. In MAGs that showed a larger than 2-log fold change, CAZyme counts specifically for Glycoside hydrolases (GH) and Carbohydrate-binding modules (CBM) for all clusters showed a high representation of the profiles with GH13, GH2 and GH3 accounting for 34.1% of all counts (Supplementary Data 23 and Fig. 5). We identified a large representation of the amylolytic (starch degrading) gene family GH13 in Hylon (counts = 88), Potato (counts = 50) and R. maize (counts = 77) treatments. As expected, GH13 was weakly represented in Avicel (counts = 19) and Inulin (counts = 29) treatments (Fig. 5). The presence of GH13 in MAGs was closely associated with CBM48, which is commonly appended to starch-degrading GH13 enzymes[30]. By aggregating the CAZyme counts of MAGs per treatment, the accessible (N. maize and R. maize) and recalcitrant starches (Hylon and Potato) showed large increases in the proportion of

GH13, GH133, GH31, GH57, GH77, CBM34 and CBM48, especially in early degraders compared to their proportions in Inulin and Avicell reflecting selection for starch-degrading species (Fig. 5 and Supplementary Fig. 9).

In total we identified several genomes, to the best of our knowledge, not previously identified as degraders of the different carbohydrate treatments (Supplementary Data 17 and Supplementary Table 3). Although six genomes were identified as associated with the degradation of cellulose, none contained any characteristic cellulose-active CAZy genes indicating multiple cross-feeders. *Collinsella aerofaciens*_J (cluster 29_1)*, Candidatus Minthovivens enterohominis* (cluster 81_1) are genomes not previously identified in NCBI and that showed a 2x-log fold increase when in the presence of inulin and harboured multiple copies of inulinases (GH32). *Bacteroides uniformis* was identified during inulin fermentation to have three copies of the GH32 (inulinase) gene and a gene encoding the inulin binding domain, CBM38. The GH32 and CBM38 genes present in this MAG were found to be organised into a single PUL in the genome of *Bacteroides uniformis* with the organisation: GH32-GH32;CBM38-unk-GH32-GH32-susD-susC, providing further evidence that these genes are likely to be involved in inulin degradation (Supplementary Data 22). *Candidatus Minthovivens*

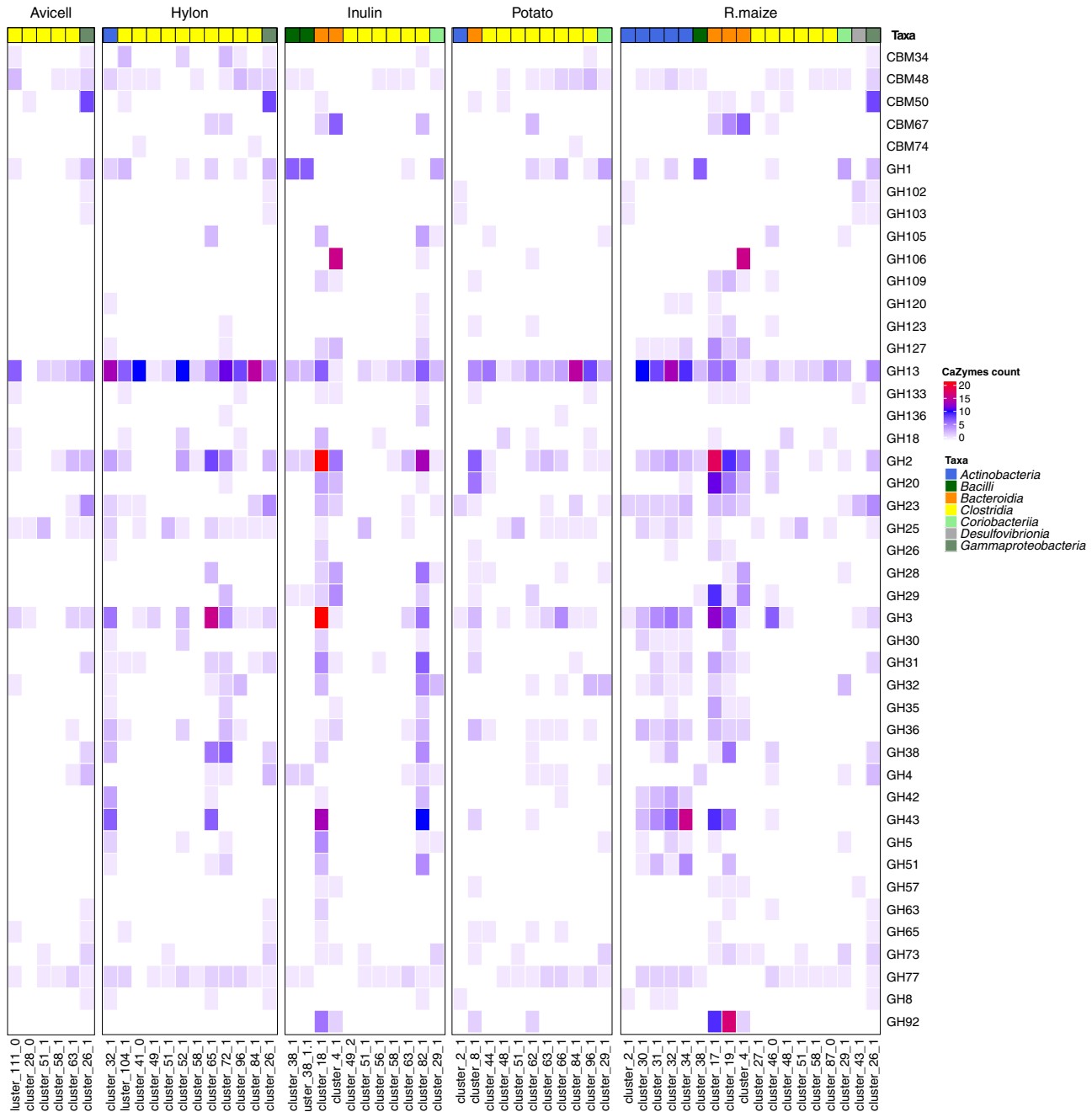

**Fig. 5 CAZyme profiles of selected-MAGs.** The colour strip represents the phylum-based taxonomy annotation. The heat map represents the number of proteins identified for each CAZy protein family.

*enterohominis* also increased in abundance early in inulin degradation, and its genome contained five copies of the GH32 gene. *R. bromii* is a well-characterised specialist on highly recalcitrant starch[31], possessing specialised starch-degrading machinery termed the 'amylosome'. Previous genome sequencing of an *R. bromii* isolate reported 15 GH13 genes[31]; 14 GH13 genes were identified in the *R. bromii* MAG assembled in this study. In the Potato treatment, a less well-characterised *Rumminococcus* species, *Candidatus Ruminococcus anthropi* with ten GH13 genes and one CBM48 gene was identified. A previously uncultured *Blautia* species, *Candidatus Blautia hennigii*, was identified possessing eight GH13 and three CBM48 genes, which increased in abundance in response to Hylon and Potato. *Blautia* species have previously been shown to

increase in abundance in response to resistant starch[32,33]. We also identified four further previously uncharacterised species that increased in abundance and had more than five GH13 genes: *Candidatus Cholicenecus caccae*, *Candidatus Eisenbergiella faecalis*, *Candidatus Enteromorpha quadrami* and *Candidatus Aphodonaster merdae*.

**Abundance of extracellular secretory CAZymes reveals substrate-specific preferences.** To further investigate the potential for substrate specificity amongst the CAZyme profiles, we calculated the changes in abundance of each CAZyme over time, by taking the abundance of each MAG and the number of copies of each CAZyme per MAG. The results of this analysis are shown in Supplementary Data 24 and Supplementary Figs. 10–15. We

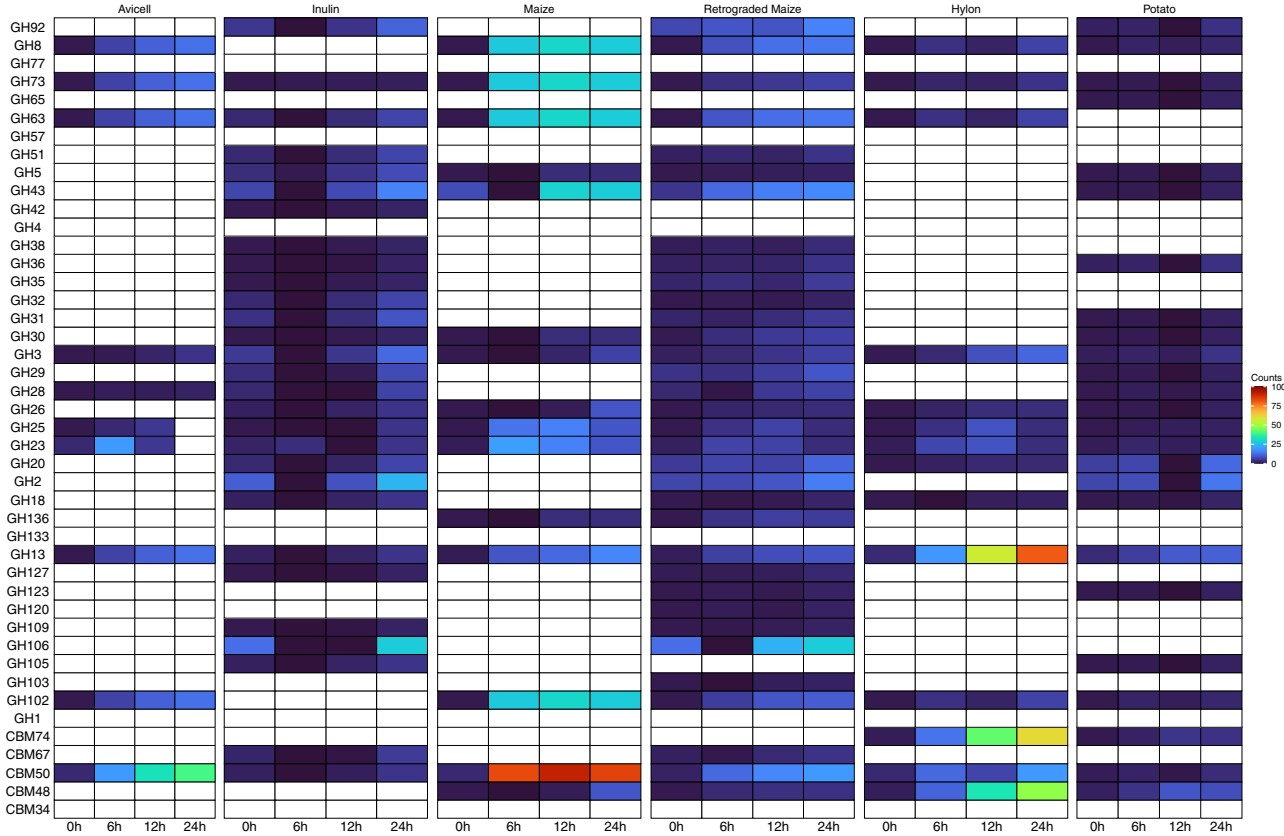

**Fig. 6 Abundance of extracellular secretory CAZymes.** A heatmap representing the changes in abundance of each CAZyme over time by taking the abundance of each MAG and the number of copies of each CAZyme per MAG. The colour strip represents the phylum-level distribution of the MAGs.

then reasoned that the CAZymes that are likely to be most directly involved in the degradation of the substrates used in this study (large molecular weight polymers with limited or no solubility), would be secretory. Therefore, we filtered the results to only include CAZymes with a signal peptide cleavage site, indicating that these enzymes will follow an extracellular secretory route (Supplementary Figs. 16–21 and Supplementary Data 25). The results reveal distinct patterns for each of the substrates depending on their structure (Fig. 6). In common with the other analyses, Avicell shows changes in relatively few CAZymes, reflecting the limited fermentability of this substrate. Inulin and R.Maize, as highly accessible substrates, show changes in a wide range of different CAZymes, while the other substrates show a more restricted range of enzymes being selected for. As discussed in the previous section, all the starch substrates show an increase in GH13 genes of the alpha-amylase family, which are key starch-degrading enzymes.

As observed in the HUMAnN3 analysis, the Hylon starch showed the greatest increase in GH13 genes relative to the other substrates. The carbohydrate-binding modules display the clearest evidence of substrate specificity in this dataset. Except Avicell and Inulin (non-starch substrates), all four of the starch substrates indicate increases in CBM48, which is a starch binding protein commonly appended to GH13 family enzymes to assist with starch degradation[30]. The two most recalcitrant starches (Hylon and Potato), but none of the other substrates, also show increases in CBM74, which may be a specific mechanism to facilitate the degradation of highly recalcitrant starches[34]. CBM74 is present in several of the MAGs, which increase in abundance in response to Hylon and Potato starches; *R. Bromii*, *Candidatus* Blautia hennigii, *Candidatus* Ruminococcus anthropic, and *Candidatus* Colihabitans norwichensis.

## Discussion

Using a hybrid assembly approach (i.e. combining NovaSeq short-read and PromethION long-read metagenomic data), we report species-level resolved taxonomic data identifying distinct changes in microbiome composition in response to different substrates. The large number of high-quality near-complete MAGs that we generated using this approach enabled us to functionally annotate the CAZymes and PULs in the MAGs and identify potential carbohydrate-degrading species. Several of these species have not previously been identified as playing a role in starch fermentation (Fig. 4 and Supplementary Table 3).

High-quality DNA for long-read sequencing was extracted using a bead beating protocol. The N50 for the PromethION reads was 4972 bp, which is comparable with another recent study using bead beating-based DNA extraction and provided adequate read lengths to be useful for assembly of MAGs[17]. A recent publication by Moss et al.[35] and associated protocol paper[36] suggested that bead beating DNA extraction protocols were unsuitable for long-read sequencing as they led to excessive shearing of DNA, and therefore, enzymatic cell lysis followed by phenol-chloroform purification were preferred to recover high molecular weight DNA. This was not reflected in our experience. The N50's obtained by Moss et al. for sequencing DNA extracted from stool samples by phenol-chloroform on the PromethION platform ranged from 1432 bp to 5205 bp, which on average was shorter than the N50 we obtained using comparable samples extracted by a bead beating protocol. This is in agreement with Bertrand et al.[14] who directly compared commercial bead beating and phenol-chloroform extraction protocols for extracting high molecular weight DNA from stool samples for MinION sequencing and found that while phenol-chloroform gave higher

molecular weights of DNA, the DNA was of low integrity compromising sequencing quality.

Hybrid assemblies allow the generation of near-complete MAGs. We found larger N50s and the longest contigs when using hybrid assemblies compared with short-read assemblies; this is in agreement with previous benchmarking data using a combined MinION and Illumina hybrid approach to sequence mock communities, human gut samples[14] and rumen gut microbiota samples[17]. This allowed us to assemble 509 MAGs across all the major phylogenetic groups (Supplementary Data 16), with representatives from ten bacterial classes and 28 families, including both Gram-positive and Gram-negative species. Bertand et al.[14] found that phenol-chloroform extractions led to the underrepresentation of 'hard to sequence' gram-positive species such as those of the genus *Bifidobacterium*. In the present study near-complete, MAG's were recovered from five different species of *Bifidobacterium*, in contrast to ref. [35] who were unable to recover *Bifidobacterium* MAG's from the PromethION data produced using their enzyme and phenol-chloroform based extraction method (although they were able to recover *Bifidobacterium* MAG's from short-read data which was obtained following a bead beating based DNA extraction of the same samples). This indicates that bead beating is necessary to obtain accurate representations of the microbial community in human stool samples. The bead beating DNA extraction protocol used in this study was also recommended by the Human Microbiome Project to avoid biases in microbiome samples[37,38].

We have provided *Candidatus* names to 70 bacterial species which do not currently have representative Latin binomial names in the GTDb database (Supplementary Note 1 and Supplementary Data 18). Our decision to provide names for these species reflects the higher quality of MAGs compared to those currently represented in the databases.

Structural diversity in substrates drives changes in microbial communities. Over the 24 h fermentation, microbial communities rapidly diverged depending on the substrate. The smallest change in community composition occurred in the Avicel treatment, as would be expected, given that Avicel was the most recalcitrant substrate evaluated, with very limited fermentability[39]. Each substrate resulted in distinct changes in microbial community composition, supporting previous findings that chemically identical but structurally-diverse starches can result in distinct changes in microbial community composition[7,8].

Changes in microbial composition are related to the ability to degrade structurally-diverse substrates. To better understand potential mechanisms driving the changes in microbial species composition in response to different substrates, we explored the CAZyme and PUL profiles of our microbial community[40].

We identified genomes that increased in abundance during either early or late stages of fermentation, suggesting that their involvement in substrate degradation was either as primary (early) or secondary (late) degraders (Fig. 4). We also identified differences in the abundance of CAZyme-encoding genes amongst species which may reflect their specialisation to specific substrates (Fig. 5). Previous studies using 16 S sequencing have identified that there are different genera of bacteria which specialise in degrading specific substrates[7,41], for example, *B. adolescentis* and *R. bromii* were previously specifically associated with high amylose maize starch and potato starch in both in vitro and in vivo studies, and we have replicated these findings[41,42]. The combination of species-level resolution and high-quality MAGs in the present study has allowed us to identify several additional species selected in response to these starches, including *Candidatus* Blautia hennigii and *Candidatus* Ruminococcus anthropic. We have highlighted that several of the species which increase in abundance in response to recalcitrant starches share a raw granular starch binding domain, CBM74.

CMB74 is a relatively recently described CBM as a domain of a raw starch-degrading amylase identified in the waste stream of a potato processing plant[34]. It has been hypothesised that CBM74 may facilitate the degradation of raw granular starches in the human gut[34], and the results of the present study would support this, as it was only found to increase in response to the recalcitrant starch substrates Hylon and Potato. It seems likely that the presence of this CBM facilitates the degradation of recalcitrant starches but is not required for more accessible starches[34]. This specificity may explain some of the differences in microbiome community composition seen in response to starches with different structures. Several of these species, for example, *Candidatus* Blautia hennigii, *Candidatus* Ruminococcus anthropic and *Candidatus* Colihabitans norwichensis, have not previously been identified as starch-degrading species due to their limited representation in databases, extending the range of species associated with starch degradation in humans.

We have demonstrated that deep long- and short-read metagenomic sequencing and hybrid assembly has great potential for studying the human gut microbiota. We identified species-level resolved changes in microbial community composition and diversity in response to carbohydrates with different structures over time, identifying the succession of species within the fermenter. To provide functional information about these species we obtained over 500 MAGs from a single human stool sample. Annotating CAZyme genes in MAGs from species enriched by fermentation of different carbohydrates allowed us to identify species specialised in the degradation of defined carbohydrates, increasing our knowledge of the range of species potentially involved in starch metabolism in the human gut.

## Methods

A schematic overview of the workflow and experimental design is displayed in Fig. 1: a human faecal sample is used in a fermentation experiment with carbohydrate substrates, using a colon model to investigate carbohydrate degradation and its effect on the gut microbiota. Samples obtained from the colon model are used for short- and long-read metagenomics sequencing. Bioinformatics analyses include hybrid assembly, genome annotation, phylogenetic analysis and CAZyme annotation to characterise changes in the microbiome for different types of carbohydrate across time. A more detailed overview of the tools and bioinformatic analyses is shown in Supplementary Fig. 1.

**Substrates**. Native maize starch (catalogue no. S4126), native potato starch (catalogue no. 2004), Avicel PH-101 (catalogue no. 11365) and chicory inulin (catalogue no. I2255) were purchased from Sigma-Aldrich, (Gillingham, UK). Hylon VII® was kindly provided as a gift by Ingredion Incorporated (Manchester, UK).

Retrograded maize starch was prepared from 40 g of native maize starch in 400 mL of deionized water. The slurry was stirred continuously at 95 °C in a water bath for 20 min. The resulting gel was cooled to room temperature for 60 min, transferred to aluminium pots (150 mL, Ampulla, Hyde UK), and stored at 4 °C for 48 h. The retrograded gel was then frozen at −80 °C for 12 h and freeze-dried (LyoDry, MechaTech Systems Ltd, Bristol, UK) for 72 h.

Each substrate (0.500 ± 0.005 g, dry weight) was weighed in sterilised fermentation bottles (100 mL) prior to the start of the experiment.

**Inoculum collection and preparation**. A single human faecal sample was obtained from one adult (≥18 years old), a free-living, healthy donor who had not taken antibiotics in the 3 months prior to donation and was free from gastrointestinal disease. Ethical approval was granted by Human Research Governance Committee at the Quadram Institute (IFR01/2015) and London—Westminster Research Ethics Committee (15/LO/2169) and the trial was registered on clinicaltrials.gov (NCT02653001). Signed informed consent was obtained from the participant prior to donation. The stool sample was collected by the participant, stored in a closed container under ambient conditions, transferred to the laboratory and prepared for inoculation within 2 hours of excretion. The faecal sample was diluted 1:10 with pre-warmed, anaerobic, sterile phosphate buffer saline (0.1 M, pH 7.4) in a double meshed stomacher bag (500 mL, Seward, Worthing, UK) and homogenised using a Stomacher 400 (Seward, Worthing, UK) at 200 rpm for two cycles, each of 60 s length.

**Batch fermentation in the colon model**. Fermentation vessels were established with media adapted from Williams et al.[43]. In brief, each vessel (100 mL) contained

an aliquot (3.0 mL) of filtered faecal slurry, 82 mL of sterilised growth medium, and one of the six substrates for experimental evaluation: native Hylon VII or native potato starch (highly recalcitrant starches); native maize starch or gelatinised, retrograded maize starch (accessible starches); Avicel PH-101 (insoluble fibre; negative control); and chicory inulin (fermentable soluble fibre; positive control). There was also a media-only control with no inoculum (blank), making a total of seven fermentation vessels.

For each fermentation vessel, the growth medium contained 76 mL of basal solution, 5 mL vitamin phosphate and sodium carbonate solution, and 1 mL reducing agent. The composition of the various solutions used in the preparation of the growth medium is described in detail in Supplementary Note 2. A single stock (7 l) of growth medium was prepared for use in all vessels. Vessel fermentations were pH controlled and maintained at pH 6.8 to 7.2 using 1 N NaOH and 1 N HCl regulated by a Fermac 260 (Electrolab Biotech, Tewkesbury, UK). A circulating water jacket maintained the vessel temperature at 37 °C. Magnetic stirring was used to keep the mixture homogenous and the vessels were continuously sparged with nitrogen (99% purity) to maintain anaerobic conditions. Samples were collected from each vessel at 0 (5 min), 6, 12 and 24 h after inoculation. The biomass from two 1.8 mL aliquots from each sample were concentrated by refrigerated centrifugation (4 °C; 10,000 × g for 10 min), the supernatant removed, and the pellets stored at −80 °C prior to bacterial enumeration and DNA extraction; one pellet was used for enumeration and one for DNA extraction.

**Bacterial cell enumeration**. All materials used for bacterial cell enumeration were purchased from Sigma-Aldrich (Gillingham, UK) unless specified otherwise. To each frozen pellet, 400 µL of Phosphate Buffered Saline (PBS) and 1100 µL of 4% paraformaldehyde were added and gently thawed at 20 °C for 10 min with gentle mixing. Once thawed, each resuspension was thoroughly mixed and incubated overnight at 4 °C for fixation to occur. The resuspensions were then centrifuged for 10 min at 8000 × g, the supernatant removed, and the residual pellet washed with 1 mL 0.1% Tween-20. This pellet then underwent two further washes in PBS to remove any residual paraformaldehyde and was then resuspended in 600 µL PBS: ethanol (1:1).

The fixed resuspensions were centrifuged for 3 min at 16,000 × g, the supernatant removed, and the pellet resuspended in 500 µL 1 mg/mL lysozyme (100 µL 1 M Tris HCl at pH 8, 100 µL 0.5 M EDTA at pH 8, 800 µL water, and 1 mg lysozyme, catalogue no. L6876) and incubated at room temperature for 10 min. After thorough mixing and centrifugation for 3 min at 16,000 × g, the supernatant was removed, and the pellet was washed with PBS. The resulting pellet was then resuspended in 150 µL of hybridisation buffer (HB, per mL: 180 µL 5 M NaCl, 20 µL 1 M Tris HCl at pH 8, 300 µL Formamide, 499 µL water, 1 µL 10% SDS), centrifuged, the supernatant removed, and the remaining pellet resuspended again in 1500 µL of HB and stored at 4 °C prior to enumeration. For bacterial enumeration, 1 µL of Invitrogen SYTO 9 (catalogue no. S34854, Thermo Fisher Scientific, Loughborough, UK) was added to 1 mL of each fixed and washed resuspension. Within 96-well plate resuspensions were diluted to 1:1000 and the bacterial populations within them enumerated using flow cytometry (Luminex Guava easyCyte 5) at wavelength of 488 nm and Guava suite software, version 3.3.

**DNA extraction**. Each pellet was resuspended in 500 µL (samples collected at 0 and 6 h) or 650 µL (samples collected at 12 and 24 h) with chilled (4 °C) nuclease-free water (Sigma-Aldrich, Gillingham, UK). The resuspensions were frozen overnight at −80 °C, thawed on ice and an aliquot (400 µL) used for bacterial genomic DNA extraction. FastDNA® Spin Kit for Soil (MP Biomedical, Solon, US) was used according to the manufacturer's instructions which included two bead beating steps of 60 s at a speed of 6.0 m/s (FastPrep24, MP Biomedical, Solon, USA). DNA concentration was determined using the Quant-iT™ dsDNA Assay Kit, high sensitivity kit (Invitrogen, Loughborough, UK) and quantified using a FLUOstar Optima plate reader (BMG Labtech, Aylesbury, UK).

**Illumina NovaSeq library preparation and sequencing**. Genomic DNA was normalised to 5 ng/µL with elution buffer (10 mM Tris HCl). A miniaturised reaction was set up using the Nextera DNA Flex Library Prep Kit (Illumina, Cambridge, UK). 0.5 µL Tagmentation Buffer 1 (TB1) was mixed with 0.5 µL Bead-Linked Transposomes (BLT) and 4.0 µL PCR-grade water in a master mix and 5 µL was added to each well of a chilled 96-well plate. About 2 µL of normalised DNA (10 ng total) was pipette-mixed with each well of tagmentation master mix and the plate heated to 55 °C for 15 min in a PCR block. A PCR master mix was made up using 4 µL kapa2G buffer, 0.4 µL dNTP's, 0.08 µL Polymerase and 4.52 µL PCR-grade water, from the Kap2G Robust PCR kit (Sigma-Aldrich, Gillingham, UK) and 9 µL added to each well in a 96-well plate. About 2 µL each of P7 and P5 of Nextera XT Index Kit v2 index primers (catalogue No. FC-131-2001 to 2004; Illumina, Cambridge, UK) were also added to each well. Finally, the 7 µL of Tagmentation mix was added and mixed. The PCR was run at 72 °C for 3 min, 95 °C for 1 min, 14 cycles of 95 °C for 10 s, 55 °C for 20 s and 72 °C for 3 min. Following the PCR reaction, the libraries from each sample were quantified using the methods described earlier and the high sensitivity Quant-iT dsDNA Assay Kit. Libraries were pooled following quantification in equal quantities. The final pool was double-SPRI size selected between 0.5 and 0.7X bead volumes using KAPA Pure Beads

(Roche, Wilmington, US). The final pool was quantified on a Qubit 3.0 instrument and run on a D5000 ScreenTape (Agilent, Waldbronn, DE) using the Agilent Tapestation 4200 to calculate the final library pool molarity. qPCR was done on an Applied Biosystems StepOne Plus machine. Samples quantified were diluted 1 in 10,000. A PCR master mix was prepared using 10 µL KAPA SYBR FAST qPCR Master Mix (2X) (Sigma-Aldrich, Gillingham, UK), 0.4 µL ROX High, 0.4 µL 10 µM forward primer, 0.4 µL 10 µM reverse primer, 4 µL template DNA, 4.8 µL PCR-grade water. The PCR programme was: 95 °C for 3 min, 40 cycles of 95 °C for 10 s, 60 °C for 30 s. Standards were made from a 10 nM stock of Phix, diluted in PCR-grade water. The standard range was 20, 2, 0.2, 0.02, 0.002, 0.0002 pmol. Samples were then sent to Novogene (Cambridge, UK) for sequencing using an Illumina NovaSeq instrument, with sample names and index combinations used. Demultiplexed FASTQ's were returned on a hard drive.

**Nanopore library preparation and PromethION sequencing**. Library preparation was performed using SQK-LSK109 (Oxford Nanopore Technologies, Oxford, UK) with barcoding kits EXP-NBD104 and EXP-NBD114. The native barcoding genomic DNA protocol by Oxford Nanopore Technologies (ONT) was followed with slight modifications. Starting material for the End-Prep/FFPE reaction was 1 µg per sample in 48 µL volume. About 3.5 µL NEBNext FFPE DNA Repair Buffer (NEB, New England Biolabs, Ipswich, USA), 3.5 µL NEB Ultra II End-prep Buffer, 3 µL NEB Ultra II End-prep Enzyme Mix and 2 µL NEBNext FFPE DNA Repair Mix (NEB) were added to the DNA (final volume 60 µL), mixed slowly by pipetting and incubated at 20 °C for 5 min and then 65 °C for 5 min. After a 1X bead wash with AMPure XP beads (Agencourt, Beckman Coulter, High Wycombe, UK), the DNA was eluted in 26 µL of nuclease-free water. About 22.5 µL of this was taken forward for native barcoding with the addition of 2.5 µL barcode and 25 µL Blunt/TA Ligase Master Mix (NEB) (final volume 50 µL). This was mixed by pipetting and incubated at room temperature for 10 min. After another 1X bead wash (as above), samples were quantified using Qubit dsDNA BR Assay Kit (Invitrogen, Loughborough, UK). In the first run, samples were equimolar pooled to a total of 900 ng in a volume of 65 µL. In the second run, samples were pooled to 1700 ng followed by a 0.4X bead wash to achieve the final volume of 65 µL. About 5 µL Adaptor Mix II (ONT), 20 µL NEBNext Quick Ligation Reaction Buffer (5X) and 10 µL Quick T4 DNA Ligase (NEB) were added (final volume 100 µL), mixed by flicking, and incubated at room temperature for 10 min. After bead washing with 50 µL of AMPure XP beads and two washes in 250 µL of Long Fragment Buffer (ONT), the library was eluted in 25 µL of Elution Buffer and quantified with Qubit dsDNA BR and TapeStation 2200 using a Genomic DNA ScreenTape (Agilent Technologies, Edinburgh, UK). About 470 ng of DNA was loaded for sequencing in the first run and 400 ng in the second run. The final loading mix was 75 µL SQB, 51 µL LB and 24 µL DNA library.

Sequencing was performed on a PromethION Beta using FLO-PRO002 PromethION Flow Cells (R9 version). The sequencing runtime was 57 h for Run 1 and 64 h for Run 2. Flow cells were refuelled with 0.5X SQB (75 µL SQB and 75 µL nuclease-free water) 40 h into both runs.

**Bioinformatics analysis**. The bioinformatics analysis was performed using default options unless specified otherwise.

*Nanopore basecalling*. Basecalling was performed using Guppy version 3.0.5 + 45c3543 (ONT) in high accuracy mode (model dna_r9.4.1_450bps_hac), and demultiplexed with qcat version 1.1.0 (Oxford Nanopore Technologies, https://github.com/nanoporetech/qcat).

*Sequence quality*. For Nanopore, sequence metrics were estimated by Nanostat version 1.1.2[44]. In total, 22 million sequences were generated with a median read length of 4500 bp and median quality of 10 (phred). Quality trimming and adaptor removal was performed using Porechop version 0.2.3 (https://github.com/rrwick/Porechop). For Illumina, quality control was done for paired-end reads using *fastp*, version 0.20.0[45] to remove adaptor sequences and filter out low-quality (phred quality <30) and short reads (length <60 bp). After quality control, the average number of reads in the samples was over 26.1 million reads, with a minimum of 9.7 million reads; the average read length was 148 bp.

*Taxonomic profiling*. Trimmed and high-quality short reads are processed using MetaPhlAn3 version 3.0.2[46] to estimate both microbial composition to species level and also the relative abundance of species from each metagenomic sample. MetaPhlAn3 uses the latest marker information dataset, CHOCOPhlAn 2019, which contains ~1 million unique clade-specific marker genes identified from ~100,000 reference genomes; this includes bacterial, archaeal, and eukaryotic genomes. Hclust2 was used to plot the hierarchical clustering of the different taxonomic profiles at each time point [https://github.com/SegataLab/hclust2]. The results of the microbial taxonomy were analysed in RStudio Version 1.1.453 (http://www.rstudio.com/).

Gene abundance was calculated using the HMP Unified Metabolic Analysis Network programme HUMAnN3[47]. The output of the HUMAnN3 pipeline was normalised to copies per million (cpm) to facilitate comparisons between samples with different read depths and regrouped into MetaCyc reaction abundances for

analysis. Fold changes were calculated between the cpm counts at time 0 h to the corresponding counts at 6, 12 and 24 h using gtools R package version 3.5.0. A threshold of 1.5x was regarded as an active metabolic gene pathway.

*Hybrid assembly*. Trimmed and high-quality Illumina reads were merged for each treatment, and then used in a short-read-only assembly using Megahit version 1.1.3[48,49]. Then OPERA-MS version 0.8.2[14] was used to combine the short-read-only assembly with high-quality long reads, to create high-quality hybrid assemblies. By combining these two technologies, OPERA-MS overcomes the issue of low-contiguity of short-read-only assemblies and the low base-pair quality of long-read-only assemblies.

*Genome binning, quality, dereplication and comparative genomics of hybrid assemblies*. The hybrid co-assemblies from Opera-MS[14] were used for binning. Here, Illumina reads for each time period were mapped to the co-assembled contigs to obtain a coverage map. Bowtie2 version 2.3.4.1 was used for mapping, and samtools to convert SAM to BAM format. MaxBin2 version 2.2.6[50] and MetaBat2 version 2.12.1[51] which uses sequence composition and coverage information, were used to bin probable genomes using default parameters. The binned genomes and co-assembled contigs were integrated into Anvi'o version 6.1[52] for manual refinement and visual inspection of problematic genomes. We used the scripts: 'anvi-interactive' to visualise the genome bins; 'anvi-run-hmms' to estimate genome completeness and contamination; 'anvi-profile' to estimate coverage and detection statistics for each sample; and 'anvi-refine' to manually refine the genomes. All scripts were run using default parameters. Additionally, DAS tool version 1.1.2[53] was used to aggregate high-quality genomes from each treatment by using single copy gene-based scores and genome quality metrics to produce a list of good-quality genomes for every treatment. CheckM version 1.0.18[54] was used on all final genomes to confirm completion and contamination scores. In general, genomes with a 'quality satisfying completeness - 5*contamination >50 score' and/or with a '>60% completion and <10% contamination' score according to CheckM, were selected for downstream analyses.

*Dereplication into representative clusters*. In order to produce a dereplicated set of genomes across all treatments, dRep version 2.5.0[55] was used. Pairwise genome comparisons or Average Nucleotide Identity (ANI) was used for clustering. dRep clusters genomes with ANIs of 97% were regarded as primary clusters. Further the genomes are clustered to an ANI of 99%. These unique genomes are regarded as secondary clusters. A representative genome is provided for each of the secondary clusters.

*Metagenomic assignment and phylogenetic analyses*. Genome bins that passed quality assessment were analysed for their closest taxonomic annotation. To assign taxonomic labels, the genome set was assigned into the microbial tree of life using GTDB version 0.3.5 and database R95 to identify the closest ancestor and obtain a putative taxonomy assignment for each genome bin. For genomes where the closest ancestor could not be determined, the Relative Evolutionary Distance (RED) to the closest ancestor and taxa names were provided to taxa not previously identified in NCBI. Using these genome bins, a phylogenetic tree was constructed using Phylophlan version 0.99 and visually inspected using iTOL version 4.3.1 and ggtree from package https://github.com/YuLab-SMU/ggtree.git. The R packages ggplot2 version 3.3.2, dplyr version 1.0.2, aplot, ggtree version 2.2.4 and inkscape version 1.0.1 were used for illustrations.

*Relative abundance of genomes*. Since co-assemblies were used for binning, it was possible to calculate the proportion of reads recruited to the MAG across all time periods for each treatment. The relative abundance provides an estimate of which time point recruited the most proportion of reads. To provide this estimate in relative terms, the value was normalised to the total number of reads that was recruited for that genome. The time 0 h sample for Avicell was not available, so a mean relative abundance from the other samples (representing the same starting inoculum) at time 0 h was used to represent 0 h for Avicell. The relative abundance scores were provided by 'anvi-summarise' (from the Anvi'o package) as relative abundance. Further, fold changes were calculated between the relative abundance at time 0 h to the corresponding relative abundance at 6, 12 and 24 h using gtools R package version 3.5.0. Fold changes provide an estimate of change in MAG abundance, which might be a result of utilisation of a particular carbohydrate. Fold changes were converted to log ratios. MAGs with a fold change of 2x (log$_2$ fold change = 1) were regarded as an active carbohydrate utiliser.

*Carbohydrate metabolism analyses*. All representative genome clusters were annotated for CAZymes using dbCAN[56]. The genome's nucleotide sequences were processed with Prodigal to predict protein sequences, and then three tools were used for automatic CAZyme annotation: (a) HMMER[57] to search against the dbCAN HMM (Hidden Markov Model) database; (b) DIAMOND[58] to search against the CAZy pre-annotated CAZyme sequence database; and (c) Hotpep[59] to search against the conserved CAZyme PPR (peptide pattern recognition) short peptide library. To improve annotation accuracy, a filtering step was used to retain only hits to CAZy families found by at least two tools. PULs were identified in

MAGs using the tool PULpy[29] with the default settings. The R packages ggplot2, dplyr, ComplexHeatmap version 2.4.3 and inkscape were used for illustrations.

**Statistics and reproducibility**. Principle Coordinate analyses using the pcoa function in the ape package version 5.3 (https://www.rdocumentation.org/packages/ape/versions/5.3) and the vegan package were used to identify differences in microbiome profiles amongst treatments using the diversity analyses. To calculate the distances between the microbiomes at different time points, the perm-disp test was performed using the betadisper function from the taxonomy profiles from vegan. Significance testing between the distances was calculated using ANOVA.

**Reporting summary**. Further information on research design is available in the Nature Research Reporting Summary linked to this article.

## Data availability
Raw read data from the PromethION and NovaSeq sequencing runs can be accessed through the NCBI SRA project number PRJNA722408 and can be accessed at http://www.ncbi.nlm.nih.gov/bioproject/722408. GenBank accession numbers for individual MAG's within this ProjectID can be found in Supplementary Data 26. Source data underlying Fig. 3b, c are provided in Supplementary Data 16.

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

## Acknowledgements

We thank Dave J. Baker for assisting with sequencing and the anonymous donor who provided faecal material for this study. We thank Dr. Judith Pell for her assistance with editing the manuscript. We acknowledge the kind assistance of Prof. Aharon Oren for checking and correcting the grammar of the protologue species names. The authors gratefully acknowledge the support of the Biotechnology and Biological Sciences Research Council (BBSRC). This research was funded by: the BBSRC Institute Strategic Programme (ISP) Food Innovation and Health BB/R012512/1 and its constituent projects (BBS/E/F/000PR10343, BS/E/F/000PR10346); the BBSRC ISP Microbes in the Food Chain BB/R012504/1 and its constituent projects (BBS/E/F/000PR10348, BBS/E/F/000PR10349, BBS/E/F/000PR10352); and the BBSRC Core Capability Grant (BB/CCG1860/1). The funders had no role in study design, data collection and analysis, decision to publish or preparation of the manuscript.

## Author contributions

All authors read and contributed to the manuscript. A.R., P.T.-R. and J.A.-J. contributed equally to this work. F.J.W. conceived and designed the study. A.R. led the preparation of the manuscript. A.A. and G.L.K. prepared the sequencing libraries and did the sequencing. A.R. and P.T.R. did the sequence and bioinformatics analysis. T.L.V. did the post-sequencing analysis. J.A.-J., K.R.C. and S.H. did the model colon experiments and DNA extractions. H.H. enumerated the bacterial cells. R.G. and M.J.P. assisted with bioinformatic analysis and taxonomic descriptions. J.O.G. provided long-read sequencing and molecular biology expertise; A.J.P. provided bioinformatics expertise; and F.J.W. provided expertise in carbohydrate structure and model colon protocols. F.J.W., J.O.G. and A.J.P. secured funding, provided management oversight and scientific direction.

## Competing interests

The authors declare no competing interests.
