## [Peer Review File · Communications Biology]

Reviewers' comments:

Reviewer #1 (Remarks to the Author):

Ravi et al use in vitro methods to characterize the microbiome response to various carbohydrate sources. This is an interesting study that will be helpful to researchers in the microbiome space as they seek to build a deeper understanding of carbohydrate utilization by gut bacteria. The sequencing and bioinformatics are carried out very nicely, allowing the authors to produce high-quality metagenome-assembled genomes and make some new observations about CAZymes and growth on different substrates.

The study is primarily limited by a lack of replicates, which precludes a statistical analysis of the results. However, the authors should be granted leeway here, because running more replicates now would likely increase the costs beyond the available funds. In future studies, I would recommend that the authors run more replicates, decrease the sequencing depth to keep costs down, then pool the replicates for assembly. Such an approach would allow them to add a statistical analysis and increase the impact of their paper, without a huge cost increase.

Prior to publication, the paper should be improved in two ways: the writing of the results section must be tidied up, and the authors should push harder to take advantage of their unique and high-quality MAGs. These are summarized in the major comments.

Major comments

****YOU CANNOT PRESENT NEW RESULTS IN THE DISCUSSION.**** The results, discussion, and methods must be revised so that all new results are reported in the results section. The discussion should place the results in a broader context of other studies, and should not reiterate the results beyond what is needed to set this up. For example, on line 275 you state "We found the greatest number and diversity of CAZyme genes were in the genomes of Bacteroidetes," but this was not reported in the text of the results, though it may have been buried in a figure or table. If a finding is important enough to re-state in the discussion, it should be written out in the results.

Overall, the results section of this manuscript reads more like a methods/results, and the discussion section reads more like a results/discussion. Methods that are not new/innovative or are not directly relevant to reporting the results should be confined to the methods section. Your paper would benefit greatly by expanding the later sections of the results, carefully stepping through the findings from your MAGs. This is the major payoff of your work, and it gets short shrift in your results!

Expansion of *E. coli* in fecal samples does NOT indicate contamination (Line 126). It is likely an important result that warrants reporting. You repeated it across 3 time points, though you have no replicates. I appreciate you running a negative control sample. It seems like a lot of your negative control consists of back-contamination from your real samples, as evidenced by the high abundance of *P. copri*. Real lab contamination with no crossover from high-abundance samples is often revealed by taxa such as *Ralstonia*, *Pelomonas*, *Bradyrhizobium*, *P. acnes*, and other taxa that would almost never be found in feces.

I know you are limited from a small sample size, but can you push a little harder to establish the link between the CAZyme content and response to substrates? Do genomes with similar CAZymes respond similarly to different substrates? Does this hold for genomes in different phyla, which might have a similar substrate response?

Can you say anything about how the carbohydrate-active genes are clustered into operons across the various genomes? It seems a shame to build these nice long contigs and then not use them for an analysis of which genes are close to each other. People who do Illumina-only sequencing are normally not able to assemble contigs long enough to reliably capture whole operons, and will be jealous that you are able to carry out such an insightful analysis.

How does the CAZyme composition for each genus correspond to the composition in reference

genomes from NCBI? Do your results reflect what is already known about the genera, or have you found things that are totally new?

The authors propose a bunch of new candidate species names. Do you have a record of prior publications that would indicate that such suggestions are adopted by the wider community of scientists? If not, I suggest you forego the introduction of candidate names. My worry is that the candidate names will muddy the taxonomic waters, and will not be adopted as real species names anyway, when the day comes that these bacteria are cultured and deposited. I am more than willing to defer to the suggestion of the editor on this issue.

I have a number of minor comments that will improve the presentation of results.

(Line 104) The study diagram in Figure 1 is very nice for giving readers an overview of the study. However, presenting all the details of the analytical workflow, including software used, may be too much for a main figure. Please consider moving some of these details into a supplemental figure to increase the impact of the main figure.

(Line 122, 149) The text says, "Error! Reference source not found." Not sure what that's about.

Figure 2 is not referenced in the text.

The heatmap in Figure 2 works well as an overview of the taxonomic composition, but does a poor job of displaying which species increase with each carbohydrate source. I strongly suggest adding a new panel to show which species increase the most for each carbohydrate source. Adding this panel will support your presentation of the results in the text.

Suggest moving Figure 3 to supplemental. These quality metrics are not surprising, but will be of interest to readers who are hoping to do something like this in their own lab. Use your main figures to show off the MAGs and the CAZyme composition.

(Line 138) You can add statistical support for this statement by running a PERMDISP test to show that the distance among samples on day 0 was much less than the distance among samples on subsequent days. This test is implemented as `betadisper()` in the `vegan` package for R. You can also manually collect the distances among samples within each day and run a linear model/ANOVA to show that the distances are smaller on day 0.

Adding a color legend to the chart would vastly improve Figure 4. Also, if the regions of the bar charts correspond to the scatter plot, how come I see a lot of orange in the bar charts but no orange points in the scatter plots?

Figure 5. It is jarring to see a phylogenetic tree that spans multiple phyla, but is rooted in the middle of the Clostridia. We know from many other studies that the true root should place the phyla into separate clades. You probably set the root to the midpoint? Suggest either (1) adding an archaeal genome as an outgroup to find an appropriate root or (2) manually setting the root to a location that places the phyla into separate clades (will be a few nodes up the tree from Bifidobacteria).

Reviewer #2 (Remarks to the Author):

In the manuscript, "Linking carbohydrate structure with function in the human gut microbiome using hybrid metagenome assemblies", the authors have performed fermentation studies with human stools samples in the presence of different types of complex sugars. While this is an excellent study, my major concern is that the authors claims about the changes of gut bacteria abundances over time, seem to be performed in only one sample.

Major Concern

Since we know that there are several different microbial compositions of a "healthy gut", it makes this author wonder if these results are repeatable with stool samples from different donors. I would suggest that the authors consider either altering their paper to focus on the new species and genera observed and their gene compositions in relation to the treatment OR repeat this experiment in 2 additional donors.

Minor Concern

The authors have missing references on lines 122 and 149.

Reviewers' comments:

Reviewer #1 (Remarks to the Author):

Ravi et al use in vitro methods to characterize the microbiome response to various carbohydrate sources. This is an interesting study that will be helpful to researchers in the microbiome space as they seek to build a deeper understanding of carbohydrate utilization by gut bacteria. The sequencing and bioinformatics are carried out very nicely, allowing the authors to produce high-quality metagenome-assembled genomes and make some new observations about CAZymes and growth on different substrates.

The study is primarily limited by a lack of replicates, which precludes a statistical analysis of the results. However, the authors should be granted leeway here, because running more replicates now would likely increase the costs beyond the available funds. In future studies, I would recommend that the authors run more replicates, decrease the sequencing depth to keep costs down, then pool the replicates for assembly. Such an approach would allow them to add a statistical analysis and increase the impact of their paper, without a huge cost increase.

Prior to publication, the paper should be improved in two ways: the writing of the results section must be tidied up, and the authors should push harder to take advantage of their unique and high-quality MAGs. These are summarized in the major comments.

We would like to thank the reviewer for their positive assessment of our work and their detailed and constructive comments. We provide answers to the individual comments point-by-point below.

Major comments

1. *****YOU CANNOT PRESENT NEW RESULTS IN THE DISCUSSION.***** *The results, discussion, and methods must be revised so that all new results are reported in the results section. The discussion should place the results in a broader context of other studies, and should not reiterate the results beyond what is needed to set this up. For example, on line 275 you state "We found the greatest number and diversity of CAZyme genes were in the genomes of Bacteroidetes," but this was not reported in the text of the results, though it may have been buried in a figure or table. If a finding is important enough to re-state in the discussion, it should be written out in the results.*

Overall, the results section of this manuscript reads more like a methods/results, and the discussion section reads more like a results/discussion. Methods that are not new/innovative or are not directly relevant to reporting the results should be confined to the methods section. Your paper would benefit greatly by expanding the later sections of the results, carefully stepping through the findings from your MAGs. This is the major payoff of your work, and it gets short shrift in your results!

We have extensively revised the manuscript in response to the reviewers' comments, moving significant sections from the results to methods which describe mainly methodological approaches. We have removed any discussion of new findings from the discussion to the results, and we have extended the description of the MAG's. As these changes are extensive, we have not included them in detail here, they are highlighted **in red** in the manuscript.

2. *Expansion of E. coli in fecal samples does NOT indicate contamination (Line 126). It is likely an*

important result that warrants reporting. You repeated it across 3 time points, though you have no replicates. I appreciate you running a negative control sample. It seems like a lot of your negative control consists of back-contamination from your real samples, as evidenced by the high abundance of *P. copri*. Real lab contamination with no crossover from high-abundance samples is often revealed by taxa such as *Ralstonia*, *Pelomonas*, *Bradyrhizobium*, *P. acnes*, and other taxa that would almost never be found in feces.

We agree that the expansion of *E. coli* needn't be a contamination since this was not identified in Time 0 for normal maize, nor was it identified in the negative control. Therefore, we have reinstated normal maize in our analyses, results and discussion. In particular, we have included n.maize sample in Figure 4 and in Supplementary Figures 2, 3 and supplementary table 10. At line 125 we have removed the description of *E.coli* as a contamination.

3. I know you are limited from a small sample size, but can you push a little harder to establish the link between the CAZyme content and response to substrates? Do genomes with similar CAZymes respond similarly to different substrates? Does this hold for genomes in different phyla, which might have a similar substrate response?

We have further established a link between the MAGs and starch treatment by two routes- taxonomy of the MAGs and CAZyme content.

By using taxonomy of the MAGs, we identified several MAGs that have been previously discovered as degraders of the different carbohydrate treatments such as the well-known starch degrading species, *Ruminococcus bromii*. *R. bromii* was identified in the most recalcitrant starch treatments i.e. Hylon, potato and R. maize treatments. *Bifidobacterium* species was identified increasing in Maize starch treatments (r.maize, n.maize and Hylon). Previous studies have characterized *Bifidobacterium* as a starch-degrading genus. The only *Bifidobacterium* species to increase in abundance in response to Hylon was *B. adolescentis*, which is known to utilise this hard-to-digest starch better than other *Bifidobacterium* species; a broader range of *Bifidobacterium* species (*B. animalis*, *B. catenulatum*, *adolescentis*, *B. longum*) increased in abundance in response to the more accessible r.maize and n.maize substrates, suggesting these species may be better adapted to more accessible starches. MAGs. In addition, in our analyses, *Bacteroides uniformis* was identified increasing from 12h during inulin fermentation and has been previously characterised as an inulin-degrading species. In addition to this, *Faecalibacterium prausnitzii* increased in abundance with inulin supplementation and has been shown to have the ability to degrade inulin when co-cultured with primary degrading species.

CAZyme and PUL content in MAGs was used to identify novel degraders and reiterate previously identified starch degraders. *Collinsella aerofaciens*_J (cluster 29_1), *Candidatus Minthovivens enterohominis* (cluster 81_1) are novel genomes that showed a 2x log -fold increase when in the presence of inulin and harboured multiple copies of inulinases (GH32). *Bacteroides uniformis* was identified during inulin fermentation had three copies of the GH32 (inulinase) gene and a gene encoding the inulin binding domain, CBM38. The GH32 and CBM38 genes present in this MAG were found to be organised into a single PUL in the genome of *Bacteroides uniformis* with the organisation; GH32-GH32;CBM38-unk-GH32-GH32-susD-susC, providing further evidence that these genes are likely to be involved in inulin degradation. In the potato treatment, a less well characterised *Ruminococcus* species, *Candidatus Ruminococcus anthropi* with ten GH13 genes and one CBM48 gene was identified. A previously uncultured *Blautia* species, *Candidatus Blautia hennigii*, was identified possessing eight GH13 and three CBM48 genes which increased in abundance in response to Hylon and potato. We also identified four further previously-uncharacterised species that increased in abundance and had more than five GH13 genes:

Candidatus Cholicenecus caccae, *Candidatus Eisenbergiella faecalis*, *Candidatus Enteromorpha quadrami* and *Candidatus Aphodonaster merdae*.

Overall, we identified a large representation of the amylolytic (starch degrading) gene family GH13 in Hylon (counts= 88), potato (counts=50) and r.maize (counts=77) treatments. As expected, GH13 was weakly represented in Avicel (counts=19) and inulin (counts=29) treatments (Figure 5).

These findings are described in the results section at Lines 188 to 211 and lines 245 to 264

4. *Can you say anything about how the carbohydrate-active genes are clustered into operons across the various genomes? It seems a shame to build these nice long contigs and then not use them for an analysis of which genes are close to each other. People who do Illumina-only sequencing are normally not able to assemble contigs long enough to reliably capture whole operons, and will be jealous that you are able to carry out such an insightful analysis.*

We have extended our analysis of the CAZyme content of our MAGs to include analysis of organisation of CAZymes into Polysaccharide Utilisation Loci (PULs) to reflect the ability of our approach to capture complete gene operons. We have inserted a new supplementary table (Supplementary Table 12) which is a table of the identified PULs in our MAGs. We have included the following text describing these results:

Line 220: We further analysed the genome organisation of the CAZyme's identified by dbCAN2 using the tool PULpy[30], which identified Polysaccharide Utilisation Loci (PULs) in a total of 21 MAGs (Supplementary Table 12). All the PULs identified were within the phylum Bacteroidetes, and the most PULs identified within a single MAG was 79 found in *Butyricimonas faecihominis*. These statistics for numbers of PULs identified are comparable to other studies published using the same tool [17].

Line 247 The GH32 and CBM38 genes present in this MAG were found to be organised into a single PUL in the genome of *Bacteroides uniformis* with the organisation; GH32-GH32;CBM38-unk-GH32-GH32-susD-susC, providing further evidence that these genes are likely to be involved in inulin degradation.

5. *How does the CAZyme composition for each genus correspond to the composition in reference genomes from NCBI? Do your results reflect what is already known about the genera, or have you found things that are totally new?*

We have carried out a direct comparison of the CAZyme content and composition for the MAGs in our study to the nearest available reference genome available in NCBI. This was done for the 37 MAGs that increased in abundance in response to different substrates and are the most relevant MAGs for the paper. Focussing on GH family genes, the overall gene count was generally similar between the MAGs and the reference genomes. We found that the GH counts per genome were higher for 9 of the MAGs, and higher in 27 of the NCBI reference genomes, with one genome where the number of GH genes was identical between MAG and NCBI reference genomes.

We note that several of the genomes for which higher GH counts were observed in the MAGs from our study compared to NCBI reference genomes were cases where the reference genomes were obtained from environmental samples rather than isolates. This suggests that for MAGs from species that have not been isolated, the approach in this paper can yield novel information. We also note that differences in GH gene abundance between the MAGs from our study and NCBI reference genomes may reflect strain level differences in GH gene content in genomes.

We have not included this information in the paper as it does not add significantly to the overall story, but we include it below for the reference.

MAG Bin ID	GH in MAG	GH in NCBI reference genome
avicell__MAXBIN__007	19	20
avicell__METABAT__39	2	3
avicell__METABAT__50	17	37
avicell__METABAT__99	20	22
avicell__METABAT__44	48	76
hylon__MAXBIN__007	20	18
hylon__METABAT__127	22	31
hylon__METABAT__16	106	108
hylon__METABAT__172	27	22
hylon__METABAT__215	14	23
hylon__METABAT__44	4	5
hylon__METABAT__88	4	8
inulin__METABAT__130	114	141
inulin__METABAT__175	4	7
inulin__METABAT__187	4	6
inulin__METABAT__83	7	8
inulin__METABAT__87	9	12
inulin__METABAT__93	27	41
inulin__METABAT__94	35	44
nmaize__METABAT__49	6	6
nmaize__METABAT__98	62	82
pstarch__METABAT__22	59	76
pstarch__METABAT__56	25	43
pstarch__METABAT__69	14	12
rmaize__MAXBIN__026	60	56
rmaize__METABAT__133	33	57
rmaize__METABAT__166	3	2
rmaize__METABAT__170	143	149
rmaize__METABAT__174	9	14
rmaize__METABAT__5	42	40
rmaize__METABAT__56	9	6
rmaize__METABAT__8	62	54
rmaize__METABAT__99	69	85
TO__MAXBIN__063	44	54
TO__METABAT__130	19	20
TO__METABAT__242	177	195
TO__METABAT__9	38	31

6. *The authors propose a bunch of new candidate species names. Do you have a record of prior publications that would indicate that such suggestions are adopted by the wider community of*

scientists? If not, I suggest you forego the introduction of candidate names. My worry is that the candidate names will muddy the taxonomic waters, and will not be adopted as real species names anyway, when the day comes that these bacteria are cultured and deposited. I am more than willing to defer to the suggestion of the editor on this issue.

We would direct the reviewer to a recent review (<https://doi.org/10.1099/ijsem.0.005000>) which provides a strong argument in favour of the use of Candidatus names. In two recent publications (<https://doi.org/10.1186/s13059-020-1947-1> and <https://doi.org/10.7717/peerj.10941>) we have proposed large numbers of Candidatus names which have been accepted for peer reviewed publication. In several cases these proposed Candidatus names have been accepted into the NCBI taxonomy and are now associated with the relevant genomes and taxa in the NCBI database (e.g. https://www.ncbi.nlm.nih.gov/nuccore/?linkname=pubmed_nuccore&from_uid=33868800).

I have a number of minor comments that will improve the presentation of results.

7. (Line 104) The study diagram in Figure 1 is very nice for giving readers an overview of the study. However, presenting all the details of the analytical workflow, including software used, may be too much for a main figure. Please consider moving some of these details into a supplemental figure to increase the impact of the main figure.

We have followed the reviewer's suggestion. Figure 1 has now been streamlined. We have included an additional Supplementary figure 1 with the full details of the workflow.

8. (Line 122, 149) The text says, "Error! Reference source not found." Not sure what that's about.

The error at line 122 and line 149 is a formatting error when converting figure hyperlinks to PDF. The hyperlink to figure 2 has now been removed, so Figure 2 is now correctly referred to in the text.

9. Figure 2 is not referenced in the text.

Figure 2 is now correctly referenced (see the response to comment 8)

10. The heatmap in Figure 2 works well as an overview of the taxonomic composition, but does a poor job of displaying which species increase with each carbohydrate source. I strongly suggest adding a new panel to show which species increase the most for each carbohydrate source. Adding this panel will support your presentation of the results in the text.

We thank the review for this suggestion. To clearly indicate species that are increasing or decreasing from a particular treatment, we calculated fold changes between the abundance of the species found at Time 0 compared to the other time points. This provided a clear overview of the species that increase the most for each carbohydrate source. The fold changes were converted to log ratios and plotted as a heatmap (Supplementary figure 2).

This Supplementary figure is referenced in the manuscript in Line 128

11. Suggest moving Figure 3 to supplemental. These quality metrics are not surprising, but will be of interest to readers who are hoping to do something like this in their own lab. Use your main figures to show off the MAGs and the CAZyme composition.

Thank you for the suggestion. This is done and is shown in Supplementary figure 6

12. (Line 138) *You can add statistical support for this statement by running a PERMDISP test to show that the distance among samples on day 0 was much less than the distance among samples on subsequent days. This test is implemented as betadisper() in the vegan package for R. You can also manually collect the distances among samples within each day and run a linear model/ANOVA to show that the distances are smaller on day 0.*

We thank the reviewer for this suggestion. We did the Permdisp test using the betadisper function. This helped us to calculate the distances between the communities between time points from all treatments. The distance of the treatments at Time 0 was close to zero while the distances at time 6h, 12h and 24h showed was greater. In addition to this, ANOVA was run using the distances between the time points. The microbiome diversity changes between time 0 to Time 6h, 12h and 24h was significant with Time 0 to time 24h with the largest distance.

Boxplot displaying the distances is shown in supplementary figure 5. The results of the ANOVA test are referenced in the text at Lines 147-151.

13. *Adding a color legend to the chart would vastly improve Figure 4. Also, if the regions of the bar charts correspond to the scatter plot, how come I see a lot of orange in the bar charts but no orange points in the scatter plots?*

We thank the reviewer for pointing out this error. The colors in Figure 3 had been accidentally switched so that there were more grey than orange points, not vice versa. Now this error has been rectified. We also included a color legend and the number of MAGs associated to each color bar.

14. *Figure 5. It is jarring to see a phylogenetic tree that spans multiple phyla, but is rooted in the middle of the Clostridia. We know from many other studies that the true root should place the phyla into separate clades. You probably set the root to the midpoint? Suggest either (1) adding an archaeal genome as an outgroup to find an appropriate root or (2) manually setting the root to a location that places the phyla into separate clades (will be a few nodes up the tree from Bifidobacteria).*

We thank the reviewer for the suggestion. We have added *Methonobrevibacter smithii* as an outgroup and rooted the tree. Now the phylogenetic tree looks more arranged along the diverse phyla. This is now named as Figure 4

Reviewer #2 (Remarks to the Author):

In the manuscript, "Linking carbohydrate structure with function in the human gut microbiome using hybrid metagenome assemblies", the authors have performed fermentation studies with human stools samples in the presence of different types of complex sugars. While this is an excellent study, my major concern is that the authors claims about the changes of gut bacteria abundances over time, seem to be performed in only one sample.

We thank the reviewer for their positive assessment of the manuscript, and we address their comments below.

Major Concern

Since we know that there are several different microbial compositions of a "healthy gut", it makes this author wonder if these results are repeatable with stool samples from different donors. I would suggest that the authors consider either altering their paper to focus on the new species and genera observed and their gene compositions in relation to the treatment OR repeat this experiment in 2 additional donors.

We agree with the reviewer and would like to refer the reviewer to the changes made in response to reviewer #1, who has raised a similar point. In response to this, we have significantly rewritten both the results and discussion in order to provide a greater focus on the gene compositions of the new genera observed in relation to the treatments which have been carried out. I refer in particular to our response to points 1,3 and 4 raised by reviewer #1

Minor Concern

The authors have missing references on lines 122 and 149.

This was due to an error introduced by a hyperlink in the document and has now been rectified.

Reviewers' comments:

Reviewer #1 (Remarks to the Author):

The researchers used an in vitro fermentation system to incubate a human fecal sample with different carbohydrate substrates. They carried out deep shotgun sequencing with long and short read platforms, which allowed them to construct high-quality metagenome-assembled genomes (MAGs). They then carried out an analysis of carbohydrate-active genes in the MAGs.

The authors have conducted a solid experiment, have dedicated substantial resources to long-read sequencing, and have done an excellent job of assembling and presenting the MAGs. However, the analysis falls short in terms of quantitatively connecting the new genomes to carbohydrate utilization and does not take full advantage of the opportunities to build relevant knowledge from a data set with so much potential. It is my sincere hope that the authors will be willing to take the analysis a few steps further and substantially increase the impact of their work.

Major comments

1. The analysis of CAZyme families, which might be the *starting point* for linking carbohydrate structure with function, is limited to only a single paragraph at the end of the results. This paper was just heating up when you ended it! Once you have the CAZyme annotations, there are so many great options for additional analysis.

a. How are the GH families correlated with the carbohydrate source across all of your genomes? How does this meet or differ from expectations?

b. Why do the majority of MAGs NOT change strongly with the carbohydrate source? If GH diversity is higher in a genome, is it able to utilize more carbohydrates from each source, and thus differ less from one source to another?

c. Quantitatively, how do the CAZyme profiles in your MAGs compare to those in existing reference genomes?

d. Are other genes (not in the CAZyme DB) correlated with the carbohydrate source?

e. Can you use your results to predict how a new or hypothetical MAG will respond to a change in carbohydrate source?

f. From previous publications, can you look up the relative amount of various carbohydrate linkages in your sources and relate this quantitatively to the GHs detected in your MAGs?

g. What fraction of carbohydrate-active genes were not accounted for by your MAGs?

Rather than a laundry list of things to do, these ideas might provide loose direction as the authors think carefully about how to take better advantage of the foundation that has been laid in their results so far.

2. The authors understandably refrain from statistical analysis, which is understandable in light of not having replicates for each carbohydrate source. Despite this limitation, there are quantitative and sometimes even statistical approaches that should be employed to reinforce the points made in the paper. For example, "the Gh profiles clustered closely together" (line 138) is testable by permutations (PERMANOVA test of distance). In the final subsection, statements about GH gene abundance should be quantitative, and may be testable depending on how you ask the question. In each subsection, I urge the authors to think about how the results could be more quantitative, and if the question might be posed in a way that a statistical comparison is appropriate.

3. In many ways, the results dwell too much on the construction and characterization of the MAGs, instead of focusing on what we might learn from them. We get an entire subsection on hybrid vs. short-read-only assemblies, but it seems obvious that hybrid assemblies will kick the pants off short-read-only methods. Then, we have another subsection on taxonomic annotation of MAGs. My advice would be to shorten the results here, so we can get to learning what the MAGs have to teach us.

4. The paper is overall short on results and long on discussion. If the authors are running into problems with length, my recommendation would be to shorten the discussion substantially.

Minor comments

1. Though it is probably covered in other papers, would you include one sentence directing the reader as to where they could find and use the MAGs from this paper in subsequent research?
2. I suggest you pull the PCoA figure (Supplementary Figure 1) into the main paper as Figure 1B.
3. I find the color scale in Figure 1A hard to read (though I have been guilty of this myself in the past). Because you have large differences in abundance, you may find that a viridis or other perceptually uniform color scale will support the visualization much better.
4. I commend your use of tools like GTDb, dbCAN2, and PhyloPhlan, and I learned a few things as I read the paper. I think you may have converted me into someone who will consider giving a Latin name to a MAG someday. Thank you for your careful methodological work.

Response to reviewers' comments for manuscript COMMSBIO-21-2064A

Reviewer #1 (Remarks to the Author):

The researchers used an in vitro fermentation system to incubate a human fecal sample with different carbohydrate substrates. They carried out deep shotgun sequencing with long and short read platforms, which allowed them to construct high-quality metagenome-assembled genomes (MAGs). They then carried out an analysis of carbohydrate-active genes in the MAGs.

The authors have conducted a solid experiment, have dedicated substantial resources to long-read sequencing, and have done an excellent job of assembling and presenting the MAGs. However, the analysis falls short in terms of quantitatively connecting the new genomes to carbohydrate utilization and does not take full advantage of the opportunities to build relevant knowledge from a data set with so much potential. It is my sincere hope that the authors will be willing to take the analysis a few steps further and substantially increase the impact of their work.

We thank the reviewer for their positive and constructive comments regarding our manuscript. In response to the reviewers suggestions, we have added several additional steps to our analysis which we believe allows us to obtain greater insights from our data.

Major comments

1. The analysis of CAZyme families, which might be the *starting point* for linking carbohydrate structure with function, is limited to only a single paragraph at the end of the results. This paper was just heating up when you ended it! Once you have the CAZyme annotations, there are so many great options for additional analysis.

a. How are the GH families correlated with the carbohydrate source across all of your genomes? How does this meet or differ from expectations?

b. Why do the majority of MAGs NOT change strongly with the carbohydrate source? If GH diversity is higher in a genome, is it able to utilize more carbohydrates from each source, and thus differ less from one source to another?

c. Quantitatively, how do the CAZyme profiles in your MAGs compare to those in existing reference genomes?

d. Are other genes (not in the CAZyme DB) correlated with the carbohydrate source?

e. Can you use your results to predict how a new or hypothetical MAG will respond to a change in carbohydrate source?

f. From previous publications, can you look up the relative amount of various carbohydrate linkages in your sources and relate this quantitatively to the GHs detected in your MAGs?

g. What fraction of carbohydrate-active genes were not accounted for by your MAGs?

Rather than a laundry list of things to do, these ideas might provide loose direction as the authors think carefully about how to take better advantage of the foundation that has been laid in their results so far.

In response to this comment, we have added several further analyses which provide more insight into our data. As suggested by the reviewer, we have not addressed this comment point-by-point, but rather have selected additional analyses based on the points raised which we believe to significantly extend the impact of the work.

In response to point 1d, we have included analysis using the HUMAnN3 pipeline which calculates gene abundances and related this quantitatively to substrate composition by analysing fold changes in gene abundance. We have included the following text to describe this in the methods (Line 587-593):

Gene abundance was calculated using the HMP Unified Metabolic Analysis Network programme HUMAnN3 [47]. The output of the HUMAnN3 pipeline was normalised to copies per million (cpm) to facilitate comparisons between samples with different read depths and the regrouped into MetaCyc reaction abundances for analysis. Foldchanges were calculated between the cpm counts at time 0h to the corresponding counts at 6h, 12h and 24h using gtools R package version 3.5.0. A threshold of 1.5x was regarded as an active metabolic gene pathway.

And in the results (Line 153-175):

Microbial metabolic gene pathways profiles were obtained using the HUMAnN3 tool (Supplementary table 3). Dynamic shifts in microbial community that also reflects shifts in abundance of metabolic gene pathways was quantified. These dynamic pathways and genes were defined as those showing at least a 1.5-fold (log₂) shift in abundance relative to baseline (time 0h) (Supplementary table 4a-4f).

With limited taxonomic shifts observed in Avicell treatment, very few genes showed distinct changes, mainly linked to cell wall remodelling and core microbial metabolism (Supplementary 4c). There was a 1.5x shift from 0h to 6h and 1.2x shift from 0h to 24h in abundance of a 6-phospho-beta-glucosidase, which can breakdown products of cellulose, possibly indicating some very limited cellulose degradation. Inulin shows broader range of enzymes involved in microbial cellular metabolism. A 2-fold increase in abundance of 1,5-anhydro-D-fructose reductase, an enzyme potentially involved in metabolism of products from inulin metabolism (Supplementary 4d). N.maize shows a non-specific pattern of gene abundances due to the high levels of E.coli (Supplementary table 4a). The substrates Potato, Hylon and R.maize all show specific changes in gene abundances related to starch degradation (Supplementary table 4b, 4e & 4f). At 24h, Hylon shows a 2-fold, and potato a 1.5-fold increase in alpha-amylase gene abundance. Potato (1.6-fold) and R.maize (1.5-fold) also show increased oligo-1,6-glucoside involved in degrading the 1,6 branch points present in Potato and R.maize (but largely absent in Hylon). In addition to starch hydrolysing enzymes, this analysis also revealed increases in abundance of a wide range of genes directly involved in acetyl-CoA and fatty acid metabolism for the Potato, R.maize and Hylon substrates. This may be linked to the production of high levels of short chain fatty acids as an endpoint of metabolism of these substrates.

In response to several of the points raised by the reviewer, specifically 1a, 1b and 1f we have included a more quantitative analysis of the CAZyme results to relate them to the species abundance for each of the substrates. Additionally, we have narrowed the focus to extracellular secretory enzymes likely to be involved in degradation of large polymeric substrates. This has yielded interesting additional results highlighting CBM74, a recently discovered raw starch binding domain, as a factor which is present in several of the not previously isolated MAG's linked to recalcitrant starch degradation. We have included supplementary figures 10 and 11, and Figure 6, as new figures showing these results. We have also included the following text in the results (Line 291-318):

Abundance of extra-cellular secretory CAZymes reveals substrate specific preferences. To further investigate the potential for substrate specificity amongst the CAZyme profiles, we calculated the changes in abundance of each CAZyme over time, by taking the abundance of each MAG and the number of copies of each CAZyme per MAG. The results of this analysis are shown in Supplemenatry

table 16 & Supplementary Figure 10a-10f. We then reasoned that the CAZymes that are likely to be most directly involved in the degradation of the substrates used in this study (large molecular weight polymers with limited or no solubility), would be secretory. Therefore, we filtered the results to only include CAZymes with a signal peptide cleavage site, indicating that these enzymes will follow an extra cellular secretory route (Supplementary Figures 11a – 11f; Supplementary table 17). The results reveal distinct patterns for each of the substrates depending on their structure (Figure 6). In common with the other analyses, Avicell shows changes in relatively few CAZymes, reflecting the limited fermentability of this substrate. Inulin and R.Maize, as highly accessible substrates, show changes in a wide range of different CAZymes, while the other substrates show a more restricted range of enzymes being selected for. As discussed in the previous section, all the starch substrates show an increase in GH13 genes, of the alpha-amylase family, which are key starch degrading enzymes.

As observed in the HUMAnN3 analysis, the Hylon starch showed the greatest increase in GH13 genes relative to the other substrates. The carbohydrate binding modules display the clearest evidence of substrate specificity in this dataset. Except Avicell and Inulin (non-starch substrates), all four of the starch substrates indicate increases in CBM48, which is a starch binding protein commonly appended to GH13 family enzymes to assist with starch degradation [30]. The two most recalcitrant starches (Hylon and Potato), but none of the other substrates, also show increases in CBM74, which may be a specific mechanism to facilitate degradation of highly recalcitrant starches [34]. CBM74 is present in several of the MAGs which increase in abundance in response to Hylon and Potato starches; R. Bromii, Candidatus Blautia hennigii, Candidatus Ruminococcus anthropic and Candidatus Colihabitans norwichensis.

And the following additional text in the discussion (387-401):

We have highlighted that several of the species which increase in abundance in response to recalcitrant starches share a raw granular starch binding domain, CBM74. CMB74 is a relatively recently described CBM as a domain of a raw starch degrading amylase identified in the waste stream of a potato processing plant [34]. It has been hypothesised that CBM74 may facilitate the degradation of raw granular starches in the human gut [34], and the results of the present study would support this, as it was only found to increase in response to the recalcitrant starch substrates Hylon and Potato. It seems likely that the presence of this CBM facilitates degradation of recalcitrant starches but is not required for more accessible starches [34]. This specificity may explain some of the differences in microbiome community composition seen in response to starches with different structures. Several of these species, for example Candidatus Blautia hennigii, Candidatus Ruminococcus anthropic and Candidatus Colihabitans norwichensis, have not previously been identified as starch degrading species due to their limited representation in databases, extending the range of species associated with starch degradation in humans.

2. The authors understandably refrain from statistical analysis, which is understandable in light of not having replicates for each carbohydrate source. Despite this limitation, there are quantitative and sometimes even statistical approaches that should be employed to reinforce the points made in the paper. For example, “the 0h profiles clustered closely together” (line 138) is testable by permutations (PERMANOVA test of distance). In the final subsection, statements about GH gene abundance should be quantitative, and may be testable depending on how you ask the question. In each subsection, I urge the authors to think about how the results could be more quantitative, and if the question might be posed in a way that a statistical comparison is appropriate.

As the reviewer points out, there is limited statistical analysis we can carry out without more replicates. The PERMANOVA analysis requested by the reviewer was carried out between timepoints

in the previous revision in the supplementary but was not explicitly stated in the main manuscript, which has now been corrected. It is not possible to test the hypothesis that the t0 timepoints cluster together due to the lack of replicates, but we can demonstrate that the t0 is statistically significantly different from all other timepoints, and that they cluster together as they are much closer to the centroid of a PCoA plot than the other timepoints (text and supplementary Figure 5) We have altered the text as follows (Line 135-152):

Dynamic shifts in the microbiome were estimated using PCoA. The distances between the microbiomes of the different treatments diverged significantly at 6h (p.value: 0.029, Permdisp ANOVA), 12h (p.value: 0.022, Permdisp ANOVA) and at 24h (p.value: 0.0099, Permdisp ANOVA). Within the time points, the profiles of R.maize and Inulin treatment profiles progressed similarly to each other, as did the Potato and Hylon treatment profiles (Supplementary Figure 3). Avicel, an insoluble fibre, showed a distinct progression of microbiome unlike the starch sources. The most distinct taxonomic change in microbial community composition in the Avicel treatment was apparent after 6h. The microbiome of N.maize progressed differently to all other treatments with E. coli taking over most variance from 6h to 24h. Figure 2b shows a PCoA plot of all the treatments excluding samples treated with N.maize to allow smaller differences between the other samples to be seen. Inverse Simpson index results followed a similar pattern for changes in diversity for most treatments, which decreased at 6h followed by a gradual increase at 12h and 24h (Supplementary Figure 4). Avicel treatment showed a different pattern of taxonomic shifts with many taxa increasing in abundance at 12h and 24h. Empirical distances between the microbiomes of different treatments between time periods showed the microbiomes were very similar at time 0 as expected (distance to centroid at 0h – 0.023) (Supplementary Figure 5).

We have aimed in the additional results included in this revision to make the outcomes more quantitative, by focussing on fold changes in abundance of genes, which allows us to make quantitative comparisons between substrates in the abundance of functional gene annotations and CAZymes (see response to question 1), which conveys the magnitude of changes observed, although statistical analysis is not possible of these results without additional replicates.

3. In many ways, the results dwell too much on the construction and characterization of the MAGs, instead of focusing on what we might learn from them. We get an entire subsection on hybrid vs. short-read-only assemblies, but it seems obvious that hybrid assemblies will kick the pants off short-read-only methods. Then, we have another subsection on taxonomic annotation of MAGs. My advice would be to shorten the results here, so we can get to learning what the MAGs have to teach us.

4. The paper is overall short on results and long on discussion. If the authors are running into problems with length, my recommendation would be to shorten the discussion substantially.

Responding to points 3 and 4 combined, in the revised manuscript we have significantly extended the results section providing two new sections describing findings arising from the relationship between the MAG's and the different substrates tested. This has significantly altered the balance of the paper in favour of results pertaining to the MAGs, rather than the construction of the MAG's themselves.

We believe there is value in retaining the sections regarding the hybrid vs short read assembly results. While it is likely that there would be significant improvements from this approach, and this is what we have demonstrated, there are to-date still very few papers demonstrating this approach. Given the relative novelty of this technical approach we believe that there is significant benefit to the community in demonstrating the benefits of hybrid assemblies in this manner. Therefore we

have retained this text in the results.

Minor comments

1. Though it is probably covered in other papers, would you include one sentence directing the reader as to where they could find and use the MAGs from this paper in subsequent research?

We are committed to open access to data and the details are covered in the additional information at the end of the manuscript. The statement of data availability is as follows and will facilitate other researchers in identifying both the MAGs and the raw sequencing data that they were generated from (line 689-694):

Availability of data and materials

Raw read data from the PromethION and NovaSeq sequencing runs can be accessed through the NCBI SRA project number PRJNA722408 and can be accessed at <https://dataview.ncbi.nlm.nih.gov/object/PRJNA722408?reviewer=ts65d8lkvj8nbv4mpfsar7sv3g>. GenBank accession numbers for individual MAG's within this ProjectID can be found in Supplementary Table 5.

2. I suggest you pull the PCoA figure (Supplementary Figure 1) into the main paper as Figure 1B.

This has been done

3. I find the color scale in Figure 1A hard to read (though I have been guilty of this myself in the past). Because you have large differences in abundance, you may find that a viridis or other perceptually uniform color scale will support the visualization much better.

The reviewer raises a good point, the colour scale used was also not colour blind friendly. We have changed all the heatmaps in the paper (main manuscript and supplementary figures) to colour scales which are perceptually uniform and colour blind friendly.

4. I commend your use of tools like GTDb, dbCAN2, and Phylophlan, and I learned a few things as I read the paper. I think you may have converted me into someone who will consider giving a Latin name to a MAG someday. Thank you for your careful methodological work.

We thank the reviewer for their kind comments

REVIEWERS' COMMENTS:

Reviewer #1 (Remarks to the Author):

The authors have addressed all my comments and substantially improved the paper. The column legends in Figure 2 and font sizes in several of the main figures may require additional tweaking by the authors, but I will leave this to be finalized in the proofs.